# Anthracyclines induce cardiotoxicity through a shared gene expression response signature

E. Renee Matthews[1], Omar D. Johnson[2], Kandace J. Horn[3], José A. Gutiérrez[1], Simon R. Powell[4], Michelle C. Ward[1]*

1 Department of Biochemistry and Molecular Biology, University of Texas Medical Branch, Galveston, Texas, United States of America, 2 Biochemistry, Cellular and Molecular Biology Graduate Program, University of Texas Medical Branch, Galveston, Texas, United States of America, 3 John Sealy School of Medicine, University of Texas Medical Branch, Galveston, Texas, United States of America, 4 Neuroscience Graduate Program, University of Texas Medical Branch, Galveston, Texas, United States of America

* miward@utmb.edu

## Abstract

TOP2 inhibitors (TOP2i) are effective drugs for breast cancer treatment. However, they can cause cardiotoxicity in some women. The most widely used TOP2i include anthracyclines (AC) Doxorubicin (DOX), Daunorubicin (DNR), Epirubicin (EPI), and the anthraquinone Mitoxantrone (MTX). It is unclear whether women would experience the same adverse effects from all drugs in this class, or if specific drugs would be preferable for certain individuals based on their cardiotoxicity risk profile. To investigate this, we studied the effects of treatment of DOX, DNR, EPI, MTX, and an unrelated monoclonal antibody Trastuzumab (TRZ) on iPSC-derived cardiomyocytes (iPSC-CMs) from six healthy females. All TOP2i induce cell death at concentrations observed in cancer patient serum, while TRZ does not. A sub-lethal dose of all TOP2i induces limited cellular stress but affects calcium handling, a function critical for cardiomyocyte contraction. TOP2i induce thousands of gene expression changes over time, giving rise to four distinct gene expression response signatures, denoted as TOP2i early-acute, early-sustained, and late response genes, and non-response genes. There is no drug- or AC-specific signature. TOP2i early response genes are enriched in chromatin regulators, which mediate AC sensitivity across breast cancer patients. However, there is increased transcriptional variability between individuals following AC treatments. To investigate potential genetic effects on response variability, we first identified a reported set of expression quantitative trait loci (eQTLs) uncovered following DOX treatment in iPSC-CMs. Indeed, DOX response eQTLs are enriched in genes that respond to all TOP2i. Next, we identified 38 genes in loci associated with AC toxicity by GWAS or TWAS. Two thirds of the genes that respond to at least one TOP2i, respond to all ACs with the same direction of effect. Our data demonstrate that TOP2i induce thousands of shared gene expression changes in cardiomyocytes, including genes near SNPs associated with inter-individual variation in response to DOX treatment and AC-induced cardiotoxicity.

**Data Availability Statement:** All RNA-seq data have been deposited in the Gene Expression Omnibus (www.ncbi.nlm.nih.gov/geo/) under accession number GSE243674. All custom

analysis scripts used for this project are available at https://github.com/mward-lab/Matthews_TOP2i_cardiotox_2023 (DOI 10.5281/zenodo.10637543) made possible by the workflowr package at https://doi.org/10.12688%2Ff1000research.20843.1.

**Funding:** This work was funded by a Cancer Prevention Research Institute of Texas (https://www.cprit.state.tx.us/) Recruitment of First-Time Faculty Award (RR190110) to M.C.W. O.D.J was supported by a Jeane B. Kempner Predoctoral Fellowship administered through UTMB (http://www.kempnerfund.org/). The funders had no role in study design, data collection and analysis, decision to publish, or preparation of the manuscript. S.R.P was supported by an NSF (http://www.nsf.gov) grant (NSF1933321) to Dr. Stefan Bossman (PI).

**Competing interests:** The authors have declared that no competing interests exist.

## Author summary

Anthracycline drugs such as Doxorubicin are effective treatments for breast cancer; however, they can cause cardiotoxicity in some women. It is unclear whether women would experience the same toxicity for all drugs in this class, or whether specific drugs would be better tolerated in specific individuals. We used an *in vitro* system of induced pluripotent stem cell-derived cardiomyocytes from six healthy females to test the effects of five breast cancer drugs on cell heath and global gene expression. We identified a strong shared cellular and gene expression response to drugs from the same class. However, there is more variation in gene expression levels between individuals following treatment with each anthracycline compared to untreated cells. We found that many genes in regions previously associated with Doxorubicin-induced cardiotoxicity in cancer patients, respond to at least two drugs in the class. This suggests that drugs in the same class induce similar effects on an individual's heart. This work contributes to our understanding of how drug response, in the context of off-target effects, varies across individuals.

## Introduction

Globally, breast cancer is the most common cancer in women [1]. 13% of women in the United States will be diagnosed with breast cancer during their lifetime [2]. While the number of deaths attributed to the disease is decreasing, and the 5-year survival rate is 90.8%, there are an estimated 3.8 million women living with breast cancer [2]. Breast cancer survivors are now more likely to suffer from secondary conditions such as cardiovascular disease (CVD) than tumor recurrence [3]. This is likely due to shared risk factors for breast cancer and CVD, and the cardiotoxic side effects induced by chemotherapeutic agents [4].

Anthracyclines (ACs) such as Doxorubicin (DOX), which is prescribed in ~32% of breast cancer patients, can cause left ventricular dysfunction and heart failure both during treatment, and years following treatment [5]. The risk of adverse cardiovascular events increases with higher doses of DOX [4]. DOX-induced congestive heart failure has been observed in 5% of patients treated with 400 mg/m$^2$ of the drug, 26% of patients treated with 550 mg/m$^2$, and 48% of patients treated with 700 mg/m$^2$ [6]. However, patients with CVD risk factors treated at low doses are also at risk for cardiotoxicity [6], suggesting inter-individual variation in response to drug treatment. Indeed, genome-wide association studies (GWAS) have implicated a handful of genetic variants in AC-induced cardiotoxicity that are close to genes including *RARG*, *SLC28A3*, *UGT1A6* and *GOLG6A2/MKRN3* [7–10].

Breast cancer survivors are more likely to suffer from heart failure and arrhythmia than ischemic heart disease, suggesting that the heart muscle itself is affected by the treatment [11]. Cardiomyocytes make up 70–85% of the heart volume [12], and are the target of DOX-induced toxicity given their high mitochondrial content and metabolic activity [13]; however these cells are challenging to obtain from humans. With the advent of iPSC technology, we are now able to acquire easily accessible cell types from blood from humans, and reprogram these cells into iPSCs, which can subsequently be differentiated into cardiomyocytes (iPSC-CMs). This *in vitro* iPSC-CM system has been shown to recapitulate the clinical effects of DOX-induced cardiotoxicity including apoptosis, DNA damage, and oxidative stress in cells from breast cancer patients treated with DOX [14]. iPSC-CMs generated from 45 healthy individuals and treated with DOX revealed hundreds of genetic variants that associate with the gene expression response to DOX treatment (eQTLs) [15]. These studies suggest that there is a genetic basis to DOX-induced cardiotoxicity, and highlight the need, and potential, for personalized medicine to reduce side effects of chemotherapeutic agents.

The precise mechanistic basis of DOX-induced cardiotoxicity is unclear. DOX intercalates into DNA forming a complex with DNA topoisomerase II (TOP2), which results in double-stranded breaks [16]. This leads to the aberrant activation of the p53 stress response pathway, mitochondrial dysfunction and apoptosis [17]. There are two TOP2 isoforms in humans–TOP2A and TOP2B. TOP2A expression is regulated in a cell cycle-dependent manner and is essential for cell division, while TOP2B contributes to transcriptional regulation in mitotic and post-mitotic cells [18]. TOP2B is essential for the cardiotoxicity observed in mice [19] and disruption of TOP2B in iPSC-CMs decreases the sensitivity to DOX [20]. In addition to inducing DNA damage, DOX has been shown to evict histones and initiate chromatin damage [21]. Indeed, in breast cancer patients, sensitivity to ACs is mediated through chromatin regulators [22]. It has been proposed that ACs, which have only chromatin-damaging activity and not DNA damage activity, do not induce cardiotoxicity [23].

The National Cancer Institute has approved tens of drugs for use in the treatment of breast cancer. This list includes ACs that are structural analogs of DOX, such as Epirubicin (EPI), which is an epimer of DOX, Daunorubicin (DNR), which differs from DOX by a hydroxyl group, and Mitoxantrone (MTX), which is an anthracenedione that is structurally related to ACs. All of these drugs are considered to be intercalating TOP2 poisons and show evidence of cardiotoxicity [24–26].

There is conflicting evidence about whether analogs such as EPI are equally or less likely to cause cardiotoxic effects than DOX [27,28]. Cardiomyopathy incidence based on pediatric cancer patient follow-up has shown that for every case of cardiomyopathy in patients treated with DOX, there are 0.8 cases for patients treated with EPI, 0.6 for DNR, and 10.5 for MTX [29]. However, clinical trials in breast cancer patients suggest that MTX is less cytotoxic than DOX [30]. Unrelated breast cancer drugs such as the HER2 receptor agonist Trastuzumab (TRZ) have also been shown to induce cardiotoxicity in as many as 6% of patients [31]; however these effects appear to be reversible [32]. It is challenging to compare drug-induced cardiotoxicity across different populations of individuals where each individual is administered a different treatment.

We thus designed a study to compare the cardiomyocyte response to five different breast cancer drugs in the same set of individuals. To do so, we established an *in vitro* model of cardiotoxicity using iPSC-CMs from a panel of six healthy female individuals and drugs that inhibit TOP2 (TOP2i) including the ACs DOX, EPI and DNR, the anthracenedione MTX, and TRZ. We collected data for cell viability, calcium handling, and global gene expression to understand the gene regulatory response to these drugs over time. This framework allowed us to identify a gene expression signature associated with the TOP2i response and gain insight into expression variability across individuals in response to different TOP2i.

## Results

We obtained iPSCs from six healthy female individuals in their twenties and thirties with no known cardiac disease or cancer diagnoses and differentiated these cells into cardiomyocytes. iPSC-CMs were metabolically selected and matured in culture for ~28 days post initiation of differentiation (See Methods). The purity of the iPSC-CM cultures was determined as the proportion of cells expressing cardiac Troponin T by flow cytometry. The median iPSC-CM purity was 97% (range 63–100%) across individuals (S1 Fig).

### TOP2-inhibiting breast cancer drugs decrease iPSC-CM viability

We studied the response of iPSC-CMs to a panel of drugs used in the treatment of breast cancer (Fig 1A). Specifically, we chose drugs belonging to AC and non-AC classes that inhibit

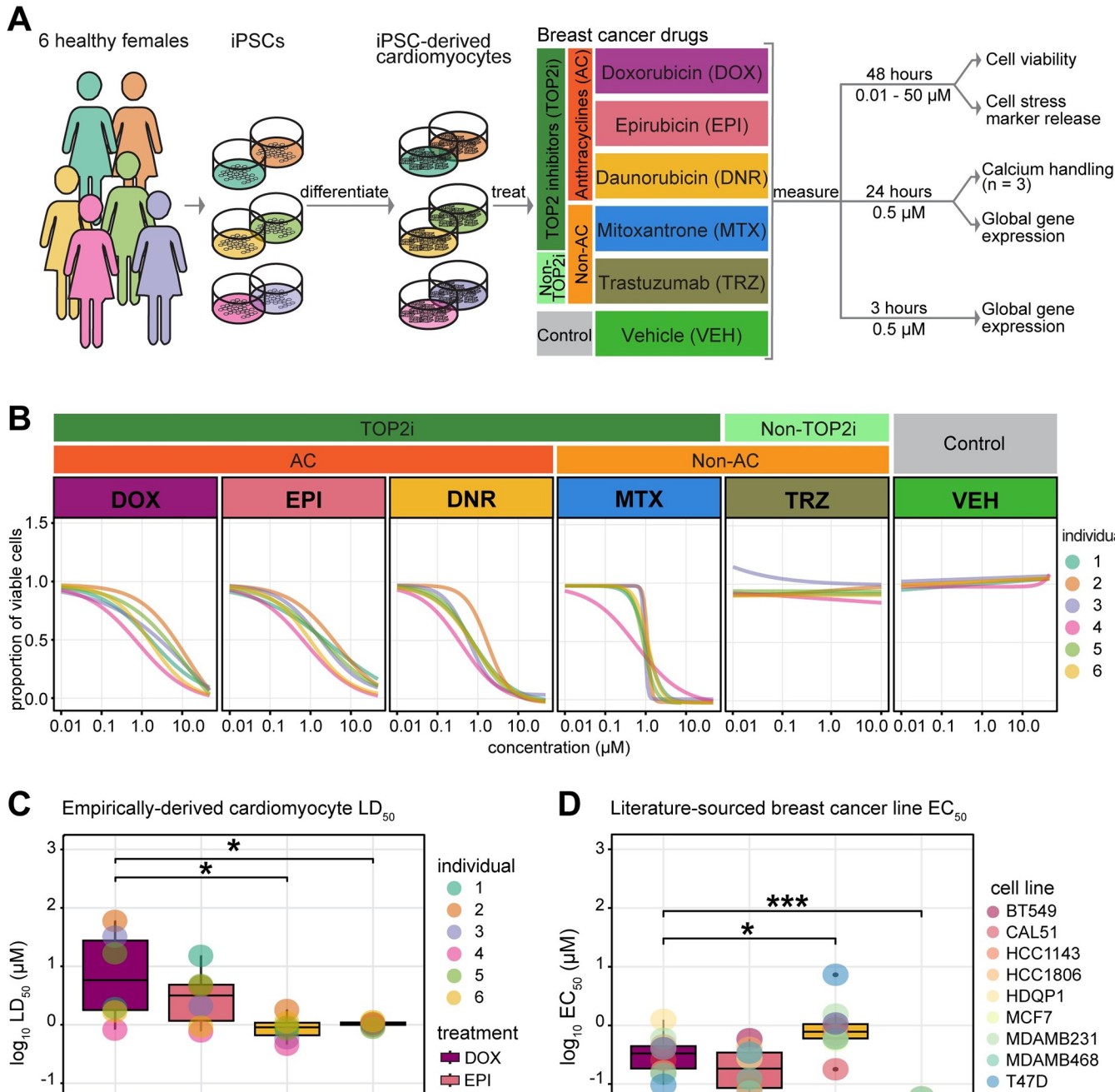

**Fig 1. TOP2i drugs affect cardiomyocyte viability in a dose-dependent manner. (A)** Experimental design of the study. iPSCs derived from six healthy women aged 21 to 32 were differentiated into cardiomyocytes (iPSC-CMs) and exposed to a panel of drugs used in the treatment of breast cancer. The response to TOP2i Doxorubicin (DOX), Epirubicin (EPI), Daunorubicin (DNR) and Mitoxantrone (MTX) is compared to the non-TOP2i Trastuzumab (TRZ) and a water vehicle (VEH). Effects on cell viability, cellular stress, cell function, and gene expression are measured. **(B)** Proportion of viable cardiomyocytes following exposure to increasing concentrations of each drug. Cell viability in each individual (colored line) was assessed following 48 hours of drug treatment. Dose-response curves were generated using a four-point log-logistic regression with the upper asymptote set to one. Each data point reflects the mean viability from two independent differentiations per individual where each is measured in quadruplicate. **(C)** $LD_{50}$ values for each drug treatment for each individual. Each value is calculated as the mean across two independent experiments. $LD_{50}$ values could not be calculated for TRZ and VEH treatments. **(D)** $EC_{50}$ values for each drug treatment in ten breast cancer cell lines were obtained from the Depmap portal (https://depmap.org) using the PRISM [33], CTRP CTD[2] [34], and GDSC2 [35] databases. Asterisk represents a statistically significant change in $LD_{50}$ (C) or $EC_{50}$ (D) values between each drug and DOX treatment (*$p < 0.05$, **$p < 0.01$, ***$p < 0.001$).

TOP2: DOX, EPI, DNR and MTX, and TRZ as an unrelated monoclonal antibody. Given the reported cytotoxic effects of several of these drugs, we first sought to measure iPSC-CM viability across six individuals following drug and vehicle (VEH) exposure at a range of drug concentrations (0.01–50 μM) for 48 hours. We observe a dose-dependent decrease in viability following DOX, EPI, DNR, and MTX treatments and no effects on viability for TRZ and VEH (Fig 1B and S1 Table). These effects are consistent across independent differentiations from the same individuals (S2 Fig), as well as across individuals (Fig 1B). Following analysis of the dose-response curves, we extracted the concentration which resulted in 50% cardiomyocyte cell death ($LD_{50}$) for each drug, across each replicate, in each individual, and calculated the median value across all six individuals. We find that the median DNR $LD_{50}$ and MTX $LD_{50}$ values are significantly lower than those for DOX (T-test; $P < 0.05$; median $LD_{50}$ in μM: DOX = 14.02, DNR = 0.98, EPI = 3.79, MTX = 0.98; Fig 1C and S2 Table).

In order to understand the cell-type specificity of the drug responses, we extracted high confidence $EC_{50}$ values following treatment with the same drugs in a panel of ten breast cancer cell lines from the PRISM, GDSC2 and CTRP databases [33–35]. Interestingly, we find that DNR is less cytotoxic in breast cancer cell lines than DOX and MTX (T-test; $P < 0.05$), and that MTX is the most cytotoxic drug in the panel (Fig 1D). This indicates that cancer drug doses tested on breast cancer cell lines may not be predictive of cardiotoxicity. To determine the physiological relevance of our model, we collated observed plasma concentrations of these drugs in patients being treated for cancer. We find the blood serum levels range from 0.002–1.73 μM, indicating that the concentrations we identified *in vitro* could be observed *in vivo* and therefore warrant further study (S3 Table).

To determine whether the effects on viability are associated with cellular stress we measured activity of secreted lactate dehydrogenase. We observe a significant inverse correlation between cell viability and cellular stress across TOP2i concentrations (DNR rho = -0.54; DOX rho = -0.35; EPI rho = -0.34; MTX rho = -0.64; $P < 0.05$) and no correlation in TRZ- and VEH-treated cells (S3 Fig and S4 Table). This suggests that samples with lower viability have a higher level of cellular stress as expected given that cardiomyocytes are post-mitotic.

Moving forward, we were interested in understanding the direct effects of these drugs on cardiomyocytes prior to the initiation of secondary effects leading to cell death. We used our dose response curve data at 48 hours (S4 Fig), together with the collated clinical serum drug concentration data, and published dose response data for DOX treatment in iPSC-CMs over time [36], to select a dose for deep characterization within the first 24 hours of treatment. We chose a treatment concentration of 0.5 μM, which is below the $LD_{50}$ calculated 48 hours post drug treatment for all drugs.

## TOP2-inhibiting breast cancer drugs affect iPSC-CM calcium handling

To elucidate the impact of cancer drugs on the calcium handling mechanism of cardiomyocytes, a fundamental determinant of cardiomyocyte contraction, we used the Fluo-4 AM fluorescent calcium indicator for real-time imaging and quantification of calcium transients. We randomly selected three individuals and treated these iPSC-CMs for 24 hours with each drug at 0.5 μM and quantified the calcium-associated fluorescence over time using spinning disc confocal microscopy (Fig 2A). We observe a significant decrease in peak amplitude for DNR, EPI and MTX compared to VEH (T-test; $P < 0.05$; Fig 2B and S5 Table), indicating decreased cytoplasmic calcium entry and therefore decreased contractility.

To estimate the rate of calcium influx and efflux from the cytosol, we examined the rising and decay slopes of the calcium transients. A reduced rate of cytosolic calcium influx during contraction was observed in DOX, EPI, and MTX samples compared to VEH (T-test;

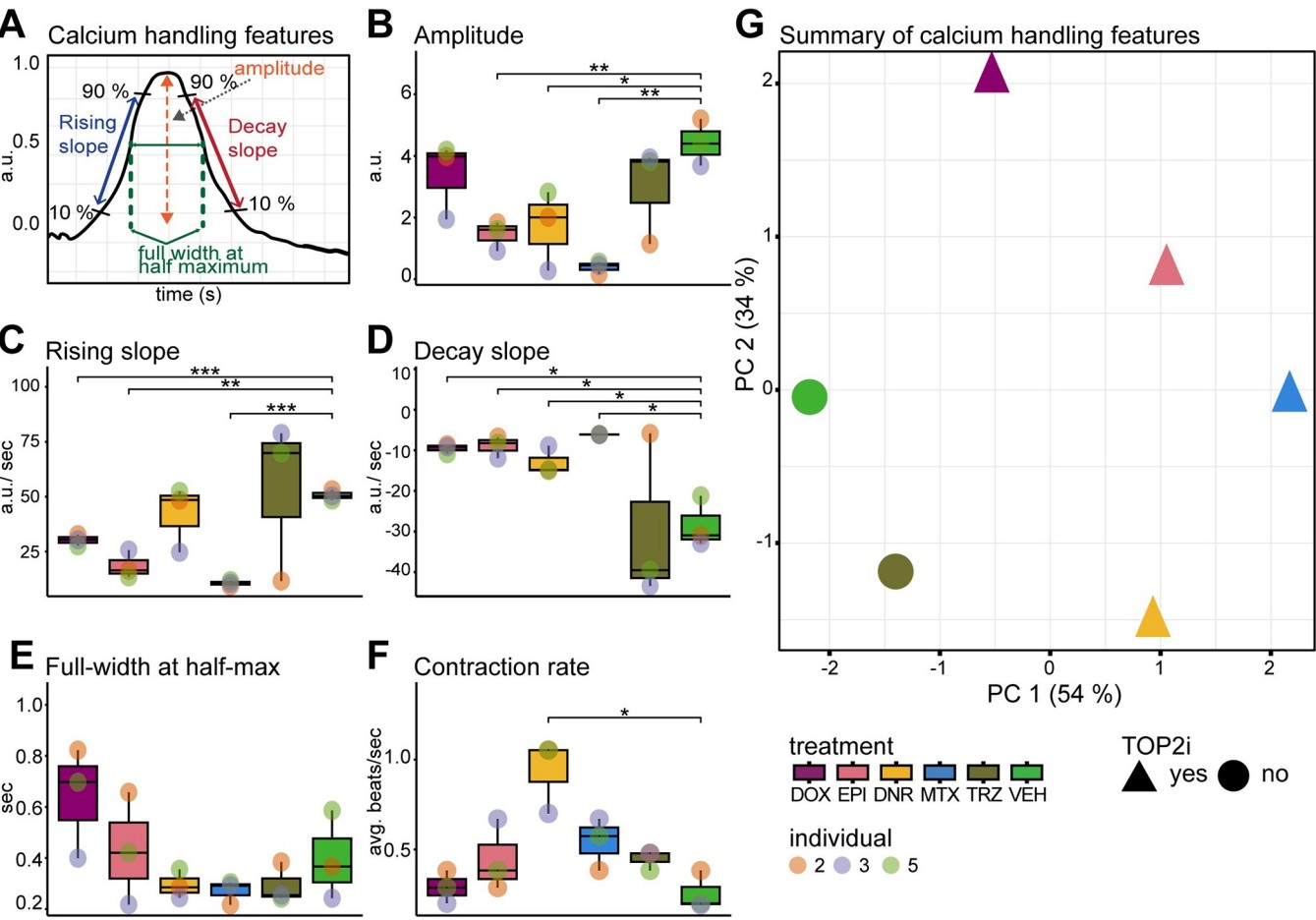

**Fig 2. Calcium dysregulation occurs in iPSC-CMs post exposure to sub-lethal concentrations of TOP2i.** **(A)** Schematic representation of the features measured to assess intracellular calcium flux in beating iPSC-CMs from three individuals treated with DOX, EPI, DNR, MTX, TRZ and VEH. Fluorescent intensity of the calcium-sensitive fluorescent dye Fluo-4 AM was measured over time by spinning disc confocal microscopy. **(B)** Mean amplitude of calcium peaks in each individual (orange dot: Individual 2, blue dot: Individual three, green dot: Individual 5) in response to DOX (mauve), EPI (pink), DNR (yellow), MTX (blue), TRZ (dark green), VEH (light green). **(C)** Mean rising slope of calcium peaks. **(D)** Mean decay slope of calcium peaks. **(E)** Mean peak width at half maximum peak height. **(F)** Mean contraction rate over ten seconds. **(G)** PCA representing five calcium flux features. TOP2i drugs are represented as triangles and non-TOP2i drugs as circles. Colors represent the specific drug treatment. Asterisk represents a statistically significant change in calcium flux measurements between drug treatments and vehicle (*$p < 0.05$, **$p < 0.01$, ***$p < 0.001$).

$P < 0.05$; Fig 2C). We observed a more gradual decay slope during the relaxation phase of contraction for DNR, DOX, EPI, and MTX samples compared to VEH (T-test; $P < 0.05$; Fig 2D). These data therefore suggest potential dysfunction in both cytosolic calcium entry and clearance. We did not observe effects of drug treatment on the duration of calcium transients, as determined by measurements of the full width of the calcium transient at half-maximum fluorescence intensity (Fig 2E). However, we did observe a significant increase in the rate of cardiomyocyte contraction in DNR-treated cells compared to VEH (T-test; $P < 0.05$; Fig 2F). Notably, contraction rate did not change in response to other drug treatments suggesting potential alterations in cardiac rhythm associated with DNR specifically.

In order to determine the overall similarity in drug effects on calcium handling, we performed principal component analysis on the data obtained for the five calcium transient features described above. PC1 which accounts for 54% of the variation, separates the VEH- and TRZ-treated samples from the TOP2i-treated samples (Fig 2G). Taken together, these findings

reveal that TOP2-inhibiting drugs at sub-lethal doses can significantly impact calcium handling in cardiomyocytes, which may contribute to cardiomyocyte dysfunction.

## AC treatments converge on a shared gene expression response over time

We were next interested in determining the gene expression changes that may lead to drug-induced effects on cardiomyocyte contraction and cell viability. To do so, we collected global gene expression measurements by RNA-seq following treatment of 80–99% pure iPSC-CMs with the five drugs at 0.5 μM, and a vehicle control, in six individuals. We assessed two time-points to capture early (3 hours) and late (24 hours) effects on gene expression. We processed the 72 high quality RNA samples in treatment and time-balanced batches (S5 Fig). Sample processing and sequencing metrics are described in S6 Table. Following sequencing, we aligned reads to the genome (S5 Fig), counted the number of reads mapping to genes and removed genes with low read counts (See Methods), leaving a final set of 14,084 expressed genes. Correlation of read counts followed by hierarchical clustering across samples identifies two distinct clusters, which primarily separates the 24-hour AC treatments from the other samples (S6 Fig). Principal component analysis reveals that PC1 accounts for 29.06% of the variation in the data, and associates with treatment time and type, while PC2, accounting for 15.76% of the variation, associates with the individual from which the samples came (Figs 3A and S7). Together, these results indicate that treatment with TOP2 inhibitors affects global gene expression in cardiomyocytes.

We next sought to identify gene expression responses to each drug treatment compared to vehicle at each timepoint using pairwise differential expression analysis. Following treatment for three hours we find tens to hundreds of differentially expressed (DE) genes at 5% FDR for the TOP2i drugs and no response to TRZ (DOX vs VEH = 19; EPI vs VEH = 210; DNR vs VEH = 532; MTX vs VEH = 75; TRZ vs VEH = 0; S8 Fig and S7–S11 Tables). The number of genes differentially expressed in response to TOP2i increases to thousands at the 24-hour time-point, while there are still no gene expression changes in response to TRZ treatment (DOX vs VEH = 6,645; EPI vs VEH = 6,328; DNR vs VEH = 7,017; MTX vs VEH = 1,115; TRZ vs VEH = 0).

Analyzing the overall magnitude of the effect of response to treatment, regardless of a p-value threshold, similarly reveals greatest responses amongst AC-treated cells at 24 hours (S9 Fig). When comparing the magnitude of the effect across all drug treatments and time by hierarchical clustering, we observe three predominant treatment clusters corresponding to the two treatment times of the TOP2i, and a TRZ treatment cluster including both timepoints (Fig 3B). We find a generally low correlation in the individual drug responses over time (DOX 3 vs 24 hour $r^2 = 0.27$; EPI $r^2 = 0.35$; DNR $r^2 = 0.28$; MTX $r^2 = 0.23$). Within the 3-hour treatment cluster, the response to DOX is moderately correlated with EPI ($r^2 = 0.68$), DNR ($r^2 = 0.61$) and MTX ($r^2 = 0.54$). The similarity between drug responses increases at the 24-hour time-point where the DOX response is highly correlated with EPI ($r^2 = 0.97$) and DNR ($r^2 = 0.98$), and moderately so with MTX ($r^2 = 0.70$).

We next compared drug responses within a timepoint by overlapping the sets of DE genes for each drug. After three hours of treatment, we find that out of 566 genes that respond to at least one TOP2i, 1% of genes respond to all four drugs and 2% respond to all three ACs (Fig 3C). After 24 hours of treatment, of the 8,188 genes that respond in at least one treatment, 11% are shared across TOP2i and 54% are shared across ACs suggesting a convergence in the response over time (Fig 3D).

While most genes respond similarly across TOP2i after 24 hours of exposure, we were interested in investigating the subset of genes that respond specifically to one drug. There are 356

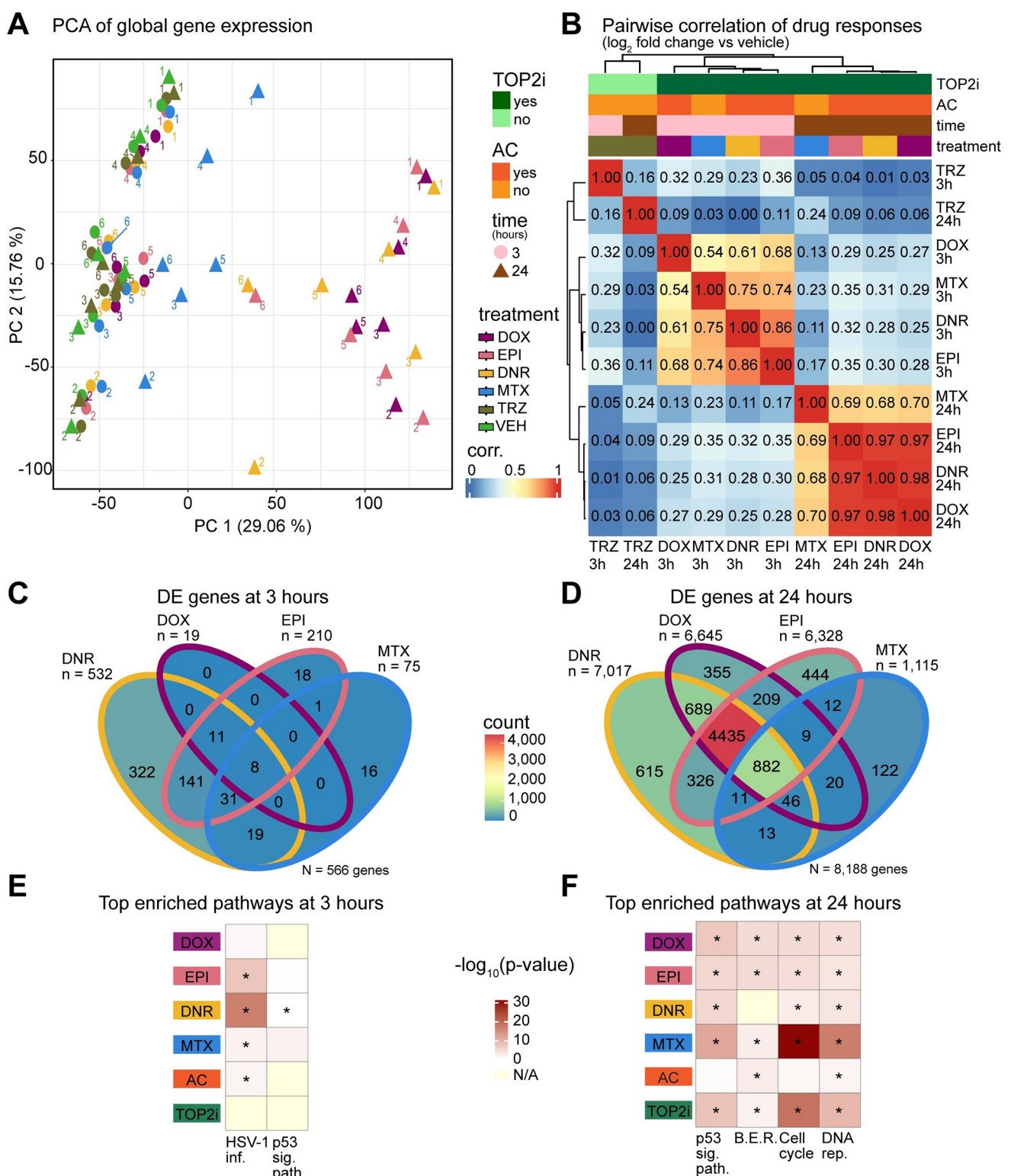

**Fig 3. TOP2i induce global gene expression changes in iPSC-CMs over 24 hours.** (A) PCA of RNA-seq-derived expression measurements ($\log_2$ cpm) across 72 samples representing six individuals (1,2,3,4,5,6), the length of exposure (three hours: circles; 24 hours: triangles), and six treatments (DOX (mauve), EPI (pink), DNR (yellow), MTX (blue), TRZ (dark green), VEH (light green)). Data is representative of 14,084 expressed genes. **(B)** Correlation between drug responses across drugs. The $\log_2$ fold change of expression between each drug treatment and VEH was calculated for all 14,084 expressed genes at each time point. These values were compared across 12 treatment-time groups by hierarchical clustering of the pairwise Pearson correlation

values. Color intensity indicates the strength of the correlation. Drug responses are colored by whether the drug is a TOP2i (TOP2i: dark green; non-TOP2i: light green), whether the drug is an AC (AC: dark orange; non-AC: light orange), and the exposure time (3 hours: pink; 24 hours: brown). **(C)** Comparison of differentially expressed genes in response to each drug treatment after three hours of exposure. n = the total number of differentially expressed genes per drug treatment. N = the total number of differentially expressed genes across drugs. **(D)** Comparison of differentially expressed genes in response to each drug treatment after 24 hours of exposure. **(E)** All biological pathways enriched amongst differentially expressed genes in response to drug treatments at three hours. Individual drug treatments as well as genes that respond to all ACs and all TOP2i are shown. KEGG pathways that are significantly enriched in at least one treatment group compared to all expressed genes are displayed (HSV-1 infection and p53 signaling pathways), where color represents the significance of the enrichment across groups and treatment groups where the pathways that are significantly enriched are represented with an asterisk. **(F)** Top biological pathways enriched amongst differentially expressed genes in response to drug treatments at 24 hours. BER: base excision repair; DNA rep.: DNA replication.

drug-specific response genes at 3 hours (DOX = 0; EPI = 18; DNR = 322; MTX = 16) and 1,536 at 24 hours (DOX = 355; EPI = 444; DNR = 615; MTX = 122) at 24 hours. However, this approach for identifying drug-specific responses with a significance threshold in each drug treatment is affected by incomplete power to detect responses in a single drug treatment. Indeed, when we investigate the drug-specific response genes we find evidence of signal in the other drug treatments, particularly in the ACs, suggesting an underestimate of the degree of sharing in response to drug treatments (S10A Fig). To identify a high confidence set of drug-specific response genes we selected genes based on the distribution of all p-values for each drug. This resulted in selection of a two-step threshold based on an adjusted p-value threshold of less than 0.01 for the drug of interest, and greater than 0.05 in all other drugs (S10B Fig). This analysis identified 104 drug-specific response genes at 3 hours (DOX = 0; EPI = 0; DNR = 100; MTX = 4; 29% of all response genes) and 305 drug-specific response genes at 24 hours (DOX = 68; EPI = 84; DNR = 112; MTX = 41; 4% of all response genes; S12 Table). Drug-specific response genes at 3 hours include the *MAFK* transcription factor for DNR, and the transcriptional regulator *ZNF547* for MTX, while 24-hour drug-specific response genes include the lncRNA gene *ZNF793-AS1* for DOX, the *SLC16A9* plasma membrane protein for EPI, the epigenetic regulator gene *SMYD4* for DNR, and the rRNA processing gene *WDR55* for MTX (S10C Fig). Together, these results suggest that drugs within the TOP2i class induce minimal drug-specific effects.

To determine the gene pathways responding to drug treatments, we performed KEGG pathway enrichment analysis on the set of genes DE in response to DOX, EPI, DNR and MTX at both timepoints and found cell cycle, p53 signaling, DNA replication, and base excision repair pathways to be amongst the most enriched pathways across drugs (adjusted $P < 0.001$ relative to all expressed genes; Fig 3E). p53 signaling and base excision repair pathways are similarly enriched amongst all drugs, while cell cycle and DNA replication genes are most enriched amongst MTX and TOP2i-shared response genes at 24 hours (Fig 3F). This set of genes includes *CDKN1A*, a p53 response gene and cell cycle regulator. These results corroborate work that investigated the response to 0.05–0.45 μM DOX over time, and identified DNA damage and cell cycle genes to be affected following treatment [37]. The only pathways enriched at three hours are p53 signaling for MTX and Herpes simplex infection for EPI, DNR and MTX. The Herpes simplex infection pathway consists of many genes related to p53 signaling and apoptosis [38], specifically C2H2-type zinc binding domain proteins suggested to be involved in the DNA damage response [39].

To determine if the stringently-identified drug-specific response genes are enriched for biological processes, we performed gene ontology analysis rather than KEGG pathway analysis, given the relatively low number of genes in this set. At three hours we find transcription-related terms to be most enriched amongst DNR- and MTX-specific response genes as well as metabolism-related terms for DNR specifically (the only two drugs that initiate a drug-specific response at this timepoint), while at 24 hours we find enrichment for terms related to cation

and calcium channel activity amongst the 68 DOX-specific response genes, and terms related to DNA replication for the 41 MTX-specific response genes (adjusted $P < 0.001$ relative to all expressed genes; S11 Fig).

We further investigated the lack of response to TRZ by using an alternative multiple testing correction approach, and again did not identify any gene expression changes. We identified only 36 genes that pass a nominal p-value threshold of 0.05 following either three or 24 hours of treatment including *PHLDA1* and *ANKRD2* (S12A and S12B Fig), Nominal TRZ DE genes at three hours are enriched in pathways related to transcriptional regulation in cancer and p53 signaling, while there are no pathways enriched amongst 24 hour response genes (S12C Fig).

Pairwise comparisons of multiple drugs and treatment times makes it challenging to determine overall trends in the data, therefore we next sought to jointly model the data to identify gene expression response signatures that may be shared across drugs or specific to a drug.

## AC-sensitive chromatin regulators are enriched amongst early response genes

To determine the appropriate model for gene expression signature selection, we used Bayesian information criterion and Akaike information criterion analysis. Using this approach we determined that there are four gene clusters that explain the major gene expression patterns across timepoints and treatments (S13A Fig). TRZ treatment did not contribute to any of the gene expression response patterns and had a probability of differential expression of less than 0.1 at both timepoints. We assigned genes to each of the four clusters based on them having a probability $> 0.5$ of belonging to that cluster (Figs 4A and S13B). Using this approach, we were able to uniquely classify 99.6% of expressed genes into one of the four clusters (S13 Table). We categorized the 7,409 genes, which do not respond to any drug at either timepoint as non-response genes (NR), the 487 genes that respond to the TOP2i drugs only after three hours of treatment as early-acute TOP2i response genes (EAR), the 5,596 genes that respond to the TOP2i drugs only after 24 hours of treatment as late TOP2i response genes (LR), and the 589 genes that respond to the TOP2i drugs at three and 24 hours as early-sustained TOP2i response genes (ESR; S14 Table). There are no clusters driven by individual drugs or the AC drug class at either timepoint. In line with the pairwise differential expression analysis, most genes that respond to drug treatment, respond specifically at the late timepoint (83%; Fig 4B). Early-sustained response genes show a heightened response in ACs at 24 hours (Fig 4C). Late response and early-sustained response clusters show a lower probability of differential expression in the MTX-treated samples at 24 hours (p = 0.3) compared to the AC drugs (p = 1), suggesting a divergence between AC and non-AC treatments over time (Fig 4C).

Using gene ontology enrichment analysis, we find that early-acute response genes and early-sustained response genes are enriched in biological processes related to transcription and the regulation of transcription compared to all expressed genes, while late response genes are enriched in terms related to mitosis and the cell cycle indicating time-dependent effects of drug treatment on cellular processes (adjusted $P < 0.001$; S13C Fig). The non-response cluster is enriched in terms related to oxidative phosphorylation and metabolism.

ACs inhibit TOP2 proteins and induce damage to both DNA and chromatin. Modified ACs, such as aclarubicin, that induce only DNA damage, are effective anti-cancer agents that show limited cardiotoxicity in mice [23], suggesting that it is the damage to chromatin that affects the heart. In large cohorts of breast cancer patients, and in breast cancer cell lines, chromatin regulator expression predicts response to AC treatment [22]. The authors of this study identified 54 chromatin regulators amongst a curated set of 404 chromatin regulators whose expression level associated with survival in breast cancer patients treated with ACs. We

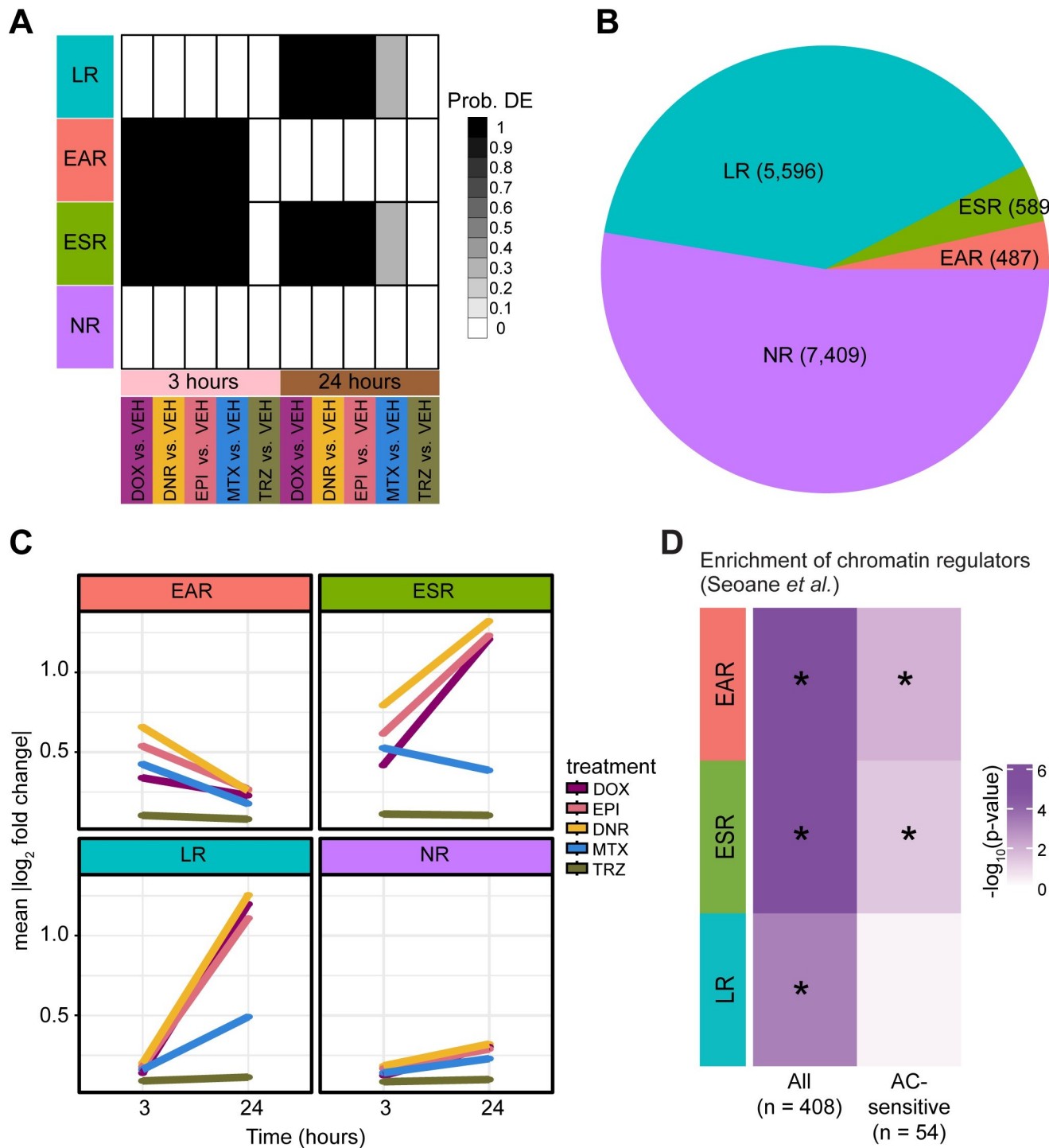

**Fig 4. Chromatin regulators that mediate breast cancer sensitivity to ACs are enriched amongst early TOP2i response genes.** (A) Gene expression motifs identified following joint modeling of test pairs. Shades of black represent posterior probabilities of genes being differentially expressed in response to each drug treatment compared to VEH at each time point. Genes are categorized based on their posterior probabilities: genes with a p > 0.5 in only 24 hour TOP2i drug treatments are designated as 'Late response genes' (LR: blue), genes with a p > 0.5 in only three hour TOP2i drug treatments are designated as 'Early-acute response genes' (EAR: red), genes with a p > 0.5 in the three and 24 hour TOP2i drug treatments are designated as 'Early-sustained response genes' (ESR: green), and genes with a p < 0.5 across all tests are designated as 'Non-response genes' (NR: purple). (B) The total number of genes that are assigned to each TOP2i response category. (C) The mean absolute log$_2$ fold change of each gene in response to each drug at each timepoint within each TOP2i response category. (D) Enrichment of chromatin regulators amongst TOP2i response gene categories compared to the non-response gene category. A curated list of chromatin regulators (n = 408) and chromatin regulators that are sensitive to ACs in breast cancer patients (n = 54) was obtained from

Seoane *et al.* [22]. Enrichment amongst response categories was calculated by a Chi-square test of proportions. Color represents the -log$_{10}$ *P* value for all tests. Significant tests are denoted with an asterisk.

therefore investigated whether these regulators respond to TOP2i in cardiomyocytes by testing for overlap with our TOP2i gene expression response clusters. All three TOP2i response gene clusters are enriched in the full set of chromatin regulators compared to genes that do not respond to any treatment (Chi-square *P* < 0.05; Fig 4D). Genes that respond early in an acute or sustained manner are also enriched for AC-sensitive chromatin regulators (including *KAT6B* for example; *P* < 0.05) compared to genes that do not respond to any treatment. This suggests that chromatin regulators involved in AC sensitivity in breast cancer patients are also involved in mediating the response in cardiomyocytes. Interestingly, the early-acute gene expression signature is enriched in terms for histone modification, histone lysine methylation, histone H3K36 methylation, histone H3K36 dimethylation, regulation of chromatin organization, and heterochromatin organization from gene ontology analysis (adjusted *P* < 0.05). These terms are not enriched amongst the early-sustained and late response genes.

## AC treatment induces transcriptional variation over time

To gain insight into inter-individual variation in drug response, we next investigated the variation in gene expression levels of all 14,084 expressed genes across individuals in response to different drug treatments at each timepoint. Mean expression levels are largely stable across treatment groups (Fig 5A and S15 Table). There is a significant reduction in the variation of gene expression levels across individuals in response to all TOP2i drugs compared to VEH at three hours, while TRZ-treated cells show no change in variation (*P* < 0.05; Fig 5B). After 24 hours of treatment, the AC drugs show increased variation in gene expression levels, while MTX and TRZ show reduced variation (*P* < 0.005; Fig 5B). Given that all samples were collected and processed in treatment-balanced batches, this suggests that there is a robust response to drug treatments shortly after exposure, but that expression diverges across individuals over time. EPI shows the greatest increase in variance followed by DOX and DNR. To determine whether the increase in AC-induced variability following treatment is evident across a larger panel of individuals, we investigated variance in response to DOX treatment across iPSC-CMs using published data from 45 individuals [15]. We find that 24 hours of treatment with 0.625 μM DOX similarly increases variability in gene expression across individuals compared to vehicle treatment (S14 Fig). We next asked how the variation across individuals changes between each drug and vehicle treatment for each gene by calculating the F statistic. We tested for correlation between the F statistic across drugs and time and observed two distinct sample clusters based on time (Fig 5C). At three hours the variability is highly correlated across all drugs (r$^2$ > 0.47), while at 24 hours the samples cluster based on whether the drugs are ACs or not. EPI-induced variance is distinct from DNR and DOX but is still highly correlated (r$^2$ > 0.52 for both) compared to MTX (r$^2$ = 0.39) and TRZ (r$^2$ = 0.28). These results imply that AC treatment induces variability in expression across individuals after 24 hours of treatment, and these effects are replicated in a DOX-focused study.

## DOX response eQTLs are enriched amongst AC response genes

We were next interested in understanding how the drug response genes relate to genes whose expression is known to vary across individuals i.e., eQTLs or eGenes. We collated the set of eGenes in left ventricle heart tissue (q-value < 0.05) identified by the GTEx consortium [40] and selected those that are expressed in our iPSC-CMs (n = 6,261). We first asked whether

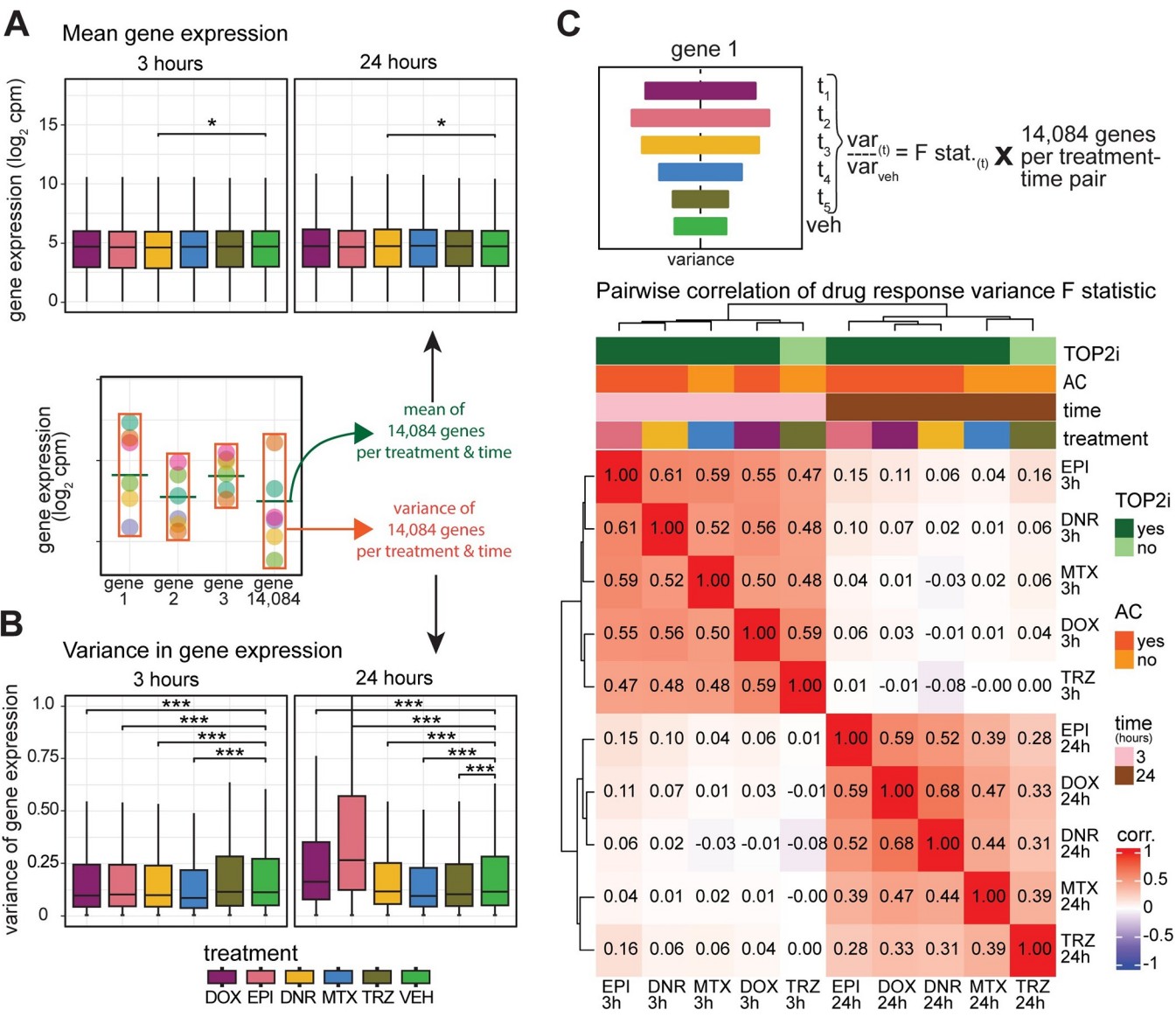

**Fig 5. AC treatments induce transcriptional variation across individuals over time.** (A) Mean expression levels of all 14,084 expressed genes across individuals for each drug treatment at each timepoint. (B) Variance of gene expression levels of all 14,084 expressed genes across individuals for each drug treatment at each timepoint. **(C)** The F-statistic was used to test for differences in the variance between each drug treatment and vehicle for all 14,084 expressed genes at each timepoint. These values were compared across 12 treatment-time groups by hierarchical clustering of the Spearman correlation values. Color intensity indicates the strength of the correlation. Drug responses are colored by whether the drug is a TOP2i (TOP2i: dark green; non-TOP2i: light green), whether the drug is an AC (AC: dark orange; non-AC: light orange), and the exposure time (3 hours: pink; 24 hours: brown). Asterisk represents a statistically significant change between drug treatment and vehicle (*$p < 0.05$, **$p < 0.005$, ***$p < 0.0005$).

genes that respond to different drug treatments are likely to vary in their expression level across individuals in the absence of treatment. We find that following 24 hours of drug treatment DOX and DNR response genes are no more likely to be an eGene in heart tissue than not, and EPI and MTX response genes are in fact depleted amongst heart eGenes (Fig 6A). This supports previous studies that suggest that in order to identify genetic effects on gene expression in disease contexts, one must study the relevant context [41]. To do so, we took advantage of a study that investigated the association between genetic variants and the response to DOX in iPSC-CMs treated with a range of DOX concentrations (0.625–5 µM)

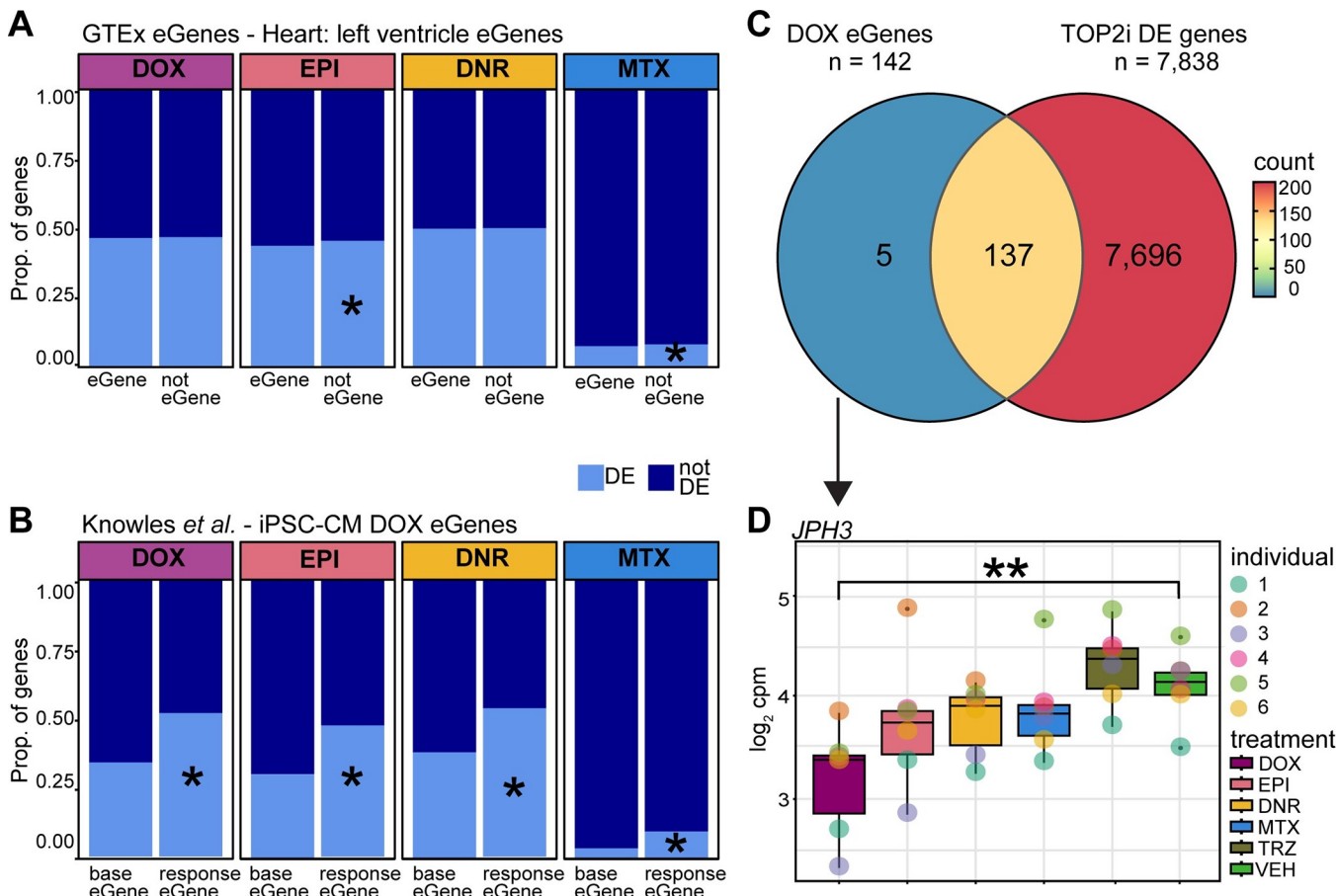

**Fig 6. DOX response eGenes are enriched amongst TOP2i response genes. (A)** Proportion of drug response genes amongst eGenes in left ventricle heart tissue. Heart eGenes (q-value < 0.05) were obtained from the GTEx database [40] and filtered to retain only those genes expressed in our data. All expressed genes that are not eGenes in heart tissue are denoted as 'not eGene'. These gene sets were compared amongst genes classified as differentially expressed in response to a particular treatment (DE: light blue) at 24 hours or not differentially expressed (not DE: dark blue). Asterisk represents drug treatments where there is a significant difference in the proportion of DE genes amongst eGenes and non-eGenes. **(B)** Proportion of drug response genes amongst DOX response eGenes in DOX-treated iPSC-CMs. eGenes identified in iPSC-CMs (base eGenes) and in response to DOX (response eGene) were obtained from Knowles *et al.* [15]. **(C)** Overlap between DOX eGenes identified by Knowles *et al.* that are responsive to DOX in our data, and genes that are differentially expressed in response to at least one other TOP2i. **(D)** Example of one of the five DOX response eGenes that responds to DOX in our cells but not to any other drug treatment. *JPH3* expression levels following 24 hours of treatment are shown. Asterisk represents drug treatments for which this gene is categorized as DE.

after 24 hours of treatment [15]. The authors identified 417 baseline eGenes in these cells, as well as 273 response eGenes i.e. eQTLs where the variant associates with the gene expression response to DOX. Reassuringly, we find that genes that respond to 0.5 μM DOX at 24 hours in our study are enriched in response eGenes compared to baseline eGenes (142 of the 273 response eGenes; Chi-square test; $P < 0.05$; Fig 6B). We also find that EPI, DNR and MTX response genes are enriched in response eGenes compared to baseline eGenes ($P < 0.05$). Of the 142 DOX response eGenes, 93% respond to DNR, 83% to EPI and 18% to MTX (14% of all DOX response genes respond to MTX), corresponding to 96% of DOX response eGenes responding to at least one TOP2i (Fig 6C). This indicates that most DOX response eGenes respond to the other AC drugs and have the potential to be DNR and EPI response eGenes. Only one DOX response eGene is categorized as one of the 68 high-stringency set of DOX-specific response genes. *JPH3*, involved in intracellular ion signaling, shows a significant response to DOX but not to any other drug (Fig 6D).

### Genes in AC toxicity-associated loci respond to ACs

We were interested in determining whether genes in AC-induced cardiotoxicity loci identified by GWAS respond to different TOP2i. To do so, we obtained genetic variant data from two GWAS [8,42] and defined a set of genes in these loci as either being the closest gene to the SNP, or an eQTL identified in any tissue, and then filtered out genes not expressed in our data (See Methods). We also included a set of four genes prioritized by a TWAS study based on AC-induced cardiotoxicity genetic data and expression data across human tissues [43]. This resulted in a set of 38 genes to interrogate for response to three and 24 hours of drug treatment. We find that only one gene responds at three hours, *GIN1*, and 22 genes respond to at least two TOP2i at 24 hours (no genes respond to a single TOP2i), 15 of which respond to all three ACs (Fig 7). AC-shared cardiotoxicity loci genes with the greatest magnitude of effect include *PELI2* and *LGALS3* that are upregulated in response to DOX, DNR and EPI, and *TNS2* and *GPSM2* that are downregulated.

We annotated each of the genes in the cardiotoxicity-associated loci with our curated set of data on response signatures, chromatin regulators, DOX-sensitive chromatin regulators, eGenes identified in heart left ventricle, and eGenes that respond to DOX in iPSC-CMs. Of the 23 genes that respond to at least one TOP2i at either timepoint, 19 are categorized as late response genes suggesting that very early response genes are not linked to toxicity. None of these genes are annotated as chromatin regulators whether AC-sensitive or not, and two genes are categorized as DOX eGenes by Knowles *et al.* (*ADCY2* and *LNPK*) [15].

Finally, we selected the set of 20 genes in cardiotoxicity-associated loci that respond to DOX (16 of these respond to all ACs) and investigated whether these genes are also DOX-responsive in a prior study [15]. We find that 18 of these genes respond with the same direction of effect (S15 Fig) indicating that these expression changes are replicated across studies.

## Discussion

While it is evident from clinical practice that DOX can induce off-target effects on the heart, we still have a poor ability to predict cardiotoxicity risk in breast cancer patients treated with DOX [44]. Similarly, it is unclear whether treatment with related AC drugs will lower the risk of cardiotoxicity or not. This uncertainty could be due to a variety of reasons including a non-standard definition of cardiotoxicity, inter-individual differences in susceptibility to a drug or class of drugs, and the fact that many variables can contribute to the phenotype. In order to understand how different breast cancer drugs affect the heart in different women, we used an *in vitro* iPSC-CM model from six healthy female individuals to control the environmental variables and measure defined endpoints including global gene expression, calcium handling, cellular stress marker release, and cell viability in response to five drugs. Using this system, we identified many effects on cardiomyocytes that are consistent following treatment with related drugs including ACs and TOP2i.

### TOP2i treatments in cardiomyocytes can inform on cardiotoxicity

iPSC-CMs have been extensively characterized as a model for studying cardiotoxicity associated with DOX [14], which has paved the way for application to other drugs such as tyrosine kinase inhibitors [45], and allowed for the development of high throughput systems to predict cardiotoxicity of novel compounds using a training set of drugs with known effects [46]. In our study, all TOP2i used in breast cancer treatment that we tested (DOX, EPI, DNR and MTX) affected cardiomyocyte viability across individuals at micromolar concentrations. These concentrations are in range of a previous *in vitro* study on AC treatments in a single individual [47], and with what has been observed clinically in cancer patients treated with

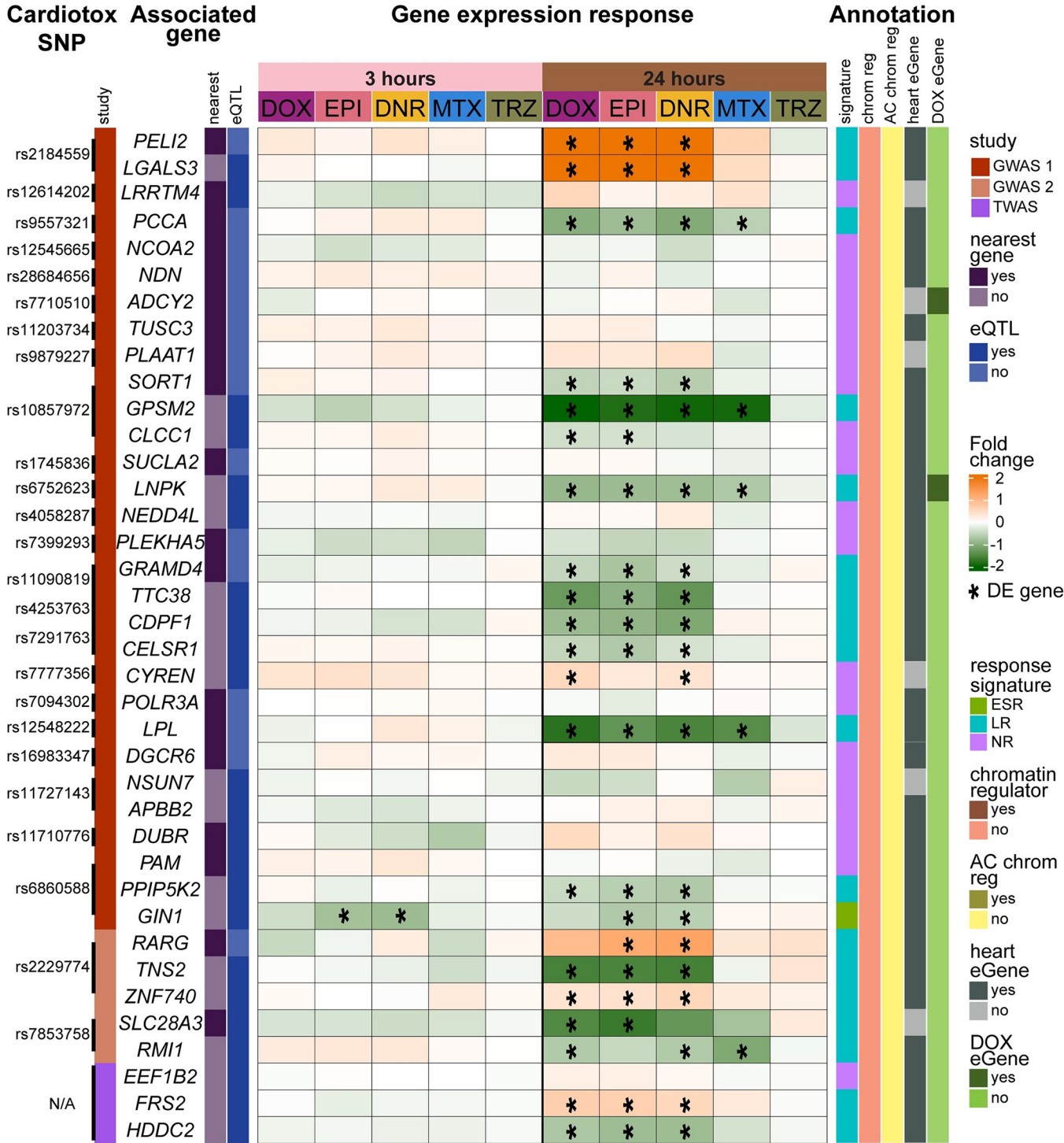

**Fig 7. Genes in AC toxicity-associated loci respond to TOP2i.** The gene expression response at three (pink) and 24 (brown) hours post drug treatment for genes in loci associated with AC-induced cardiotoxicity. The log₂ fold change for each drug compared to vehicle is shown. Genes that are classified as differentially expressed in response to a drug treatment are represented with an asterisk. GWAS loci were integrated from two GWAS studies indicated by: dark red [8], and light red [42], and one TWAS study in purple [43]. SNPs were associated with genes through either the nearest annotated gene (dark purple) or by being an eQTL in any tissue (dark blue). Each gene is annotated by their 1) response signature (ESR: Early sustained response; green, LR: Late response; light blue, and NR: Non-response: light purple), 2) chromatin regulator status: (yes: puce, no: pink), and AC-sensitive chromatin regulator status: (yes: dark yellow, no: yellow) [22], 3) left ventricle heart eGene status (yes: dark grey, no: light grey) [40], and 4) DOX response eGene status (yes: dark green; no: light green) [15].

these drugs (S3 Table). Interestingly, we observed DNR to be relatively less toxic and MTX to be more toxic in breast cancer cell lines than iPSC-CMs indicating differences in toxicities associated with cancerous and non-cancerous cells. Using sub-lethal doses of the drugs, we observed effects of TOP2i on multiple features related to calcium handling, which can be inferred to affect cardiomyocyte contraction. This suggests that these drugs affect basic cardiomyocyte function prior to leading to cell death.

## TOP2i treatments at sub-lethal concentrations induce a shared gene expression signature

We selected a sub-lethal dose of 0.5 μM for each drug to further characterize their effects on cardiomyocytes over the course of 24 hours. We did so in order to capture primary drug responses and not secondary effects as the cells succumb to stress. Indeed, it has been shown that treating iPSC-CMs with ACs at these doses for 48 hours induces the expression of death receptors [36], and the initiation of necroptosis pathways [47].

We found TOP2i to induce hundreds of gene expression changes following three hours of treatment, and thousands following 24 hours of treatment. After 24 hours, gene expression changes are enriched in pathways related to p53 signaling, the cell cycle, DNA replication and base excision repair. These gene expression changes are shared across TOP2i and can be characterized as TOP2i early-acute response genes, early-sustained and late response genes. Clinical guidelines have suggested that EPI may be better tolerated than DOX and DNR given the higher maximum recommended cumulative dose (DOX: 400–450 mg/m$^2$, EPI: 900 mg/m$^2$, DNR: 400–550 mg/m$^2$) [47]. However, we observed similar effects between EPI and DOX and DNR at this dose. A prior study has indicated that treating iPSC-CMs with a range of DOX concentrations from 0.625–5 μM for 24 hours yields several different gene expression response clusters across concentrations, and many of these effects are non-linear [15]. It is therefore possible that the gene expression changes we observe in response to TOP2i may diverge if different concentrations of drug are used.

We found that the majority of gene expression changes are shared amongst ACs with only hundreds out of thousands of response genes responding specifically to a single AC. The increased early response to DNR at three hours is likely due to the fact that the rate of DNR uptake into iPSC-CMs is approximately twice as fast as that of DOX or EPI because of its relative lipophilicity [47]. We do not see enrichment of DNR-specific pathways in our transcriptomic data at three hours–these pathways are the same as all enriched pathways in DNR response genes related to transcriptional regulation. The same is true for MTX-specific response genes at three and 24 hours. However, interestingly, genes that respond specifically to DOX at 24 hours are enriched in pathways related to calcium handling and the ryanodine-sensitive calcium release channel, unlike all DOX response genes. It is well established that DOX treatment affects calcium handling in cardiomyocytes through a variety of mechanisms [48]. DOX can bind directly to the ryanodine receptor in its closed state and can increase binding to ryanodine in a calcium-dependent manner [49]. It could therefore be the case that DOX, and not EPI or DNR, is able to associate with the ryanodine receptor leading to some drug-specific effects.

## TRZ treatment does not affect viability, calcium handling or gene expression

We did not find TRZ to have an effect on any of the phenotypes that we measured at the time-points and concentration that we used (0.5 μM) in our panel of six individuals. We did not find effects on cell viability after 48 hours of treatment at a range of concentrations up to

10 μM. This is in line with a previous study that showed that treating iPSC-CMs from breast cancer patients with 0.5 μM of TRZ for seven days does not affect viability [50]. This study does identify effects on calcium handling and gene expression after this long treatment period. We did not identify any gene expression changes in response to TRZ using our pairwise linear model or joint model across drugs following 24 hours of treatment. The aforementioned study suggests that gene expression changes induced by TRZ do not occur immediately post treatment. Indeed, a study investigating the response to 0.687 μM TRZ after 48 hours of treatment similarly did not identify any DE genes following multiple testing correction (517 genes show evidence for DE at a nominal p-value threshold of 0.05) [51]. Two of the four genes this study highlights as being downregulated by TRZ, *PHLDA1* and *SLC6A6*, are similarly two of the 36 genes downregulated in our data at a nominal p-value. Our lack of a robust gene expression response within 24 hours is therefore in line with other molecular studies of TRZ treatment in iPSC-CMs.

TRZ -induced cardiotoxicity is reported to occur in a subset of breast cancer patients [31], and to be reversible unlike DOX-induced cardiotoxicity [32] suggesting different molecular mechanisms between these drugs. It could be the case that TRZ-induced cardiotoxicity is mediated through post-transcriptional mechanisms that occur over time for example. It has also been suggested that the combination of AC and TRZ treatment, as is prescribed clinically, is more cardiotoxic than either treatment alone [52], and reduces resistance to other stress [53]. Determining transcriptional responses to combination treatments would be a next step to investigate.

## Inter-individual variation in expression increases following AC treatment

In addition to measuring mean gene expression changes in response to treatment, we also investigated the variability in response to drugs across individuals. We found a decrease in variation following three hours of treatment with all drugs, followed by an increase in variability in the AC treatments only following 24 hours of treatment. We replicated these results using data from a study investigating transcriptional changes to 24 hours of DOX treatment in iPSC-CMs from 45 individuals [15]. Variability in gene expression between individuals is similarly dynamic in the course of development where cells in a pluripotent state have lower levels of inter-individual variation than cells progressing towards a differentiated state [54,55]. In plants, highly variable genes are often environmental response genes [56]. The gene expression variability data is mirrored by the cell viability analysis where inter-individual variability in $LD_{50}$ values from MTX-treated samples is lower than AC-treated samples.

## Inter-individual susceptibility to cardiotoxicity

We began to gain insight into inter-individual susceptibility to cardiotoxicity by integrating our response genes from the five drug treatments with data connecting genes to 1) AC sensitivity in patients, 2) genes whose expression under genetic control is altered specifically following DOX treatment, and 3) genetic variants associated with cardiotoxicity. Chromatin regulators mediate sensitivity to DOX when used in the treatment of breast cancer [22]. We find that the 54 AC-sensitive chromatin regulators are enriched amongst the early TOP2i response genes suggesting that chromatin regulators may be important in inter-individual differences in susceptibility to cardiotoxicity across drugs. Intriguingly, 1,089 genes that are variable in response to DOX at a nominal p-value are enriched for pathways related to histone H3 deacetylation compared to genes that are not classified as variable. This suggests that larger studies, which include more individuals are warranted.

To specifically investigate genes whose expression has been linked to genetic variants in the context of DOX exposure, we retrieved the set of response eGenes from a study of iPSC-CMs by Knowles *et al.* [15]. We found that 96% of response eGenes respond to at least one TOP2i in our study, suggesting that they have the potential to be eGenes in response to the other drugs, and that the overall response to ACs may be similar for each individual.

Several GWAS have been performed in an attempt to identify genetic variants associated with AC-induced toxicity. These studies have often not identified genetic variants that meet the accepted threshold for genome-wide significance, and that are reproducible, possibly due to small sample sizes and heterogeneity in the definition of the cardiotoxicity phenotype. A study of ~3,000 breast cancer patients identified nine independent loci associated ($P < 5 \times 10^{-5}$) with AC-induced congestive heart failure [8]. The top SNP, which replicated in two cohorts, is in a GR binding site implying gene regulatory effects. It is located within 20 kb of the *GOLG6A2 and MKRN3* genes that are not expressed in our data. An earlier study of 280 patients treated for childhood cancer identified a single SNP ($P < 5.9 \times 10^{-8}$) in the *RARG* gene associated with cardiotoxicity in the original cohort and two replication cohorts [7]. The *RARG* gene product has been suggested to influence cardiotoxicity by binding to the *TOP2B* promoter and repressing its activity. The risk SNP is associated with decreased binding, and higher expression of TOP2B consistent with increased susceptibility to AC toxicity. To further validate the association, three individuals with risk and non-risk alleles were re-recruited from the original GWAS study in order to generate iPSCs. The effects of the SNP were recapitulated in an iPSC-CM model of DOX toxicity in these individuals [57]. Independent genetic manipulation of this susceptibility variant decreases DOX-induced toxicity in iPSC-CMs [58], and does so through the DNA damage response [59]. This SNP and gene have therefore been the best characterized cardiotoxicity-associated locus. The SNP is present in ~15% of the population [60], and may therefore contribute to cardiotoxicity risk in many individuals. Additional GWAS studies using pediatric cases have also replicated the association between cardiotoxicity and SNPs in the *SLC28A3* gene ($P = 1.9 \times 10^{-5}$) and the *UGT1A6* gene [9,10]. These initial studies, that show evidence for replication, led to the recommendation that these three SNPs be tested in childhood cancer patients being treated with ACs [42]. These recommendations state that it is unclear whether changing the type of AC will yield lower cardiotoxic effects.

Our study design allowed us to test whether different AC drugs have different cardiotoxic effects in cardiomyocytes. We therefore investigated the gene expression response to TOP2i in loci associated with cardiotoxicity [8,42,43]. Given that many GWAS-associated SNPs are in non-coding regions and have gene regulatory effects, we investigated both genes closest to the cardiotoxicity-associated SNPs, as well as genes that are associated with the SNP through eQTL analysis. This approach allowed the identification of potential novel target genes. For example, we found that *ZNF740* and *TNS2* in the *RARG* locus respond to all three ACs, and the *RMI1* gene in the *SLC28A3* locus responds to DOX, DNR, and MTX, suggesting that they too may contribute to the cardiotoxicity observed. *TNS2* is a focal adhesion molecule that binds to actin filaments. *ZNF740* is a transcriptional regulator whose DNA binding motifs are associated with enhancer usage in heart failure suggesting that this gene may indeed play a role in cardiotoxicity [61]. *RMI1* facilitates double-stranded break repair and heterodimerizes with TOP3A that has topoisomerase activity in both mitochondria and the nucleus, again suggesting a mechanism behind the association [62,63]. From the adult AC-induced cardiotoxicity GWAS study, our analysis revealed strong effects on the expression of genes including *PELI2* and *LGALS3*. These genes could have clinical implications. For example, *LGALS3*, which encodes for galectin-3 has been suggested as an early-stage circulating biomarker for various diseases including heart disease [64,65]. More broadly we found that genes responsive to DOX after 24 hours of treatment are often responsive to all other ACs suggesting similar

cardiotoxicity across these drugs, and similar mechanisms of action (16/20 genes). Notably there are no DOX-responsive genes in these loci following three hours of treatment suggesting that genes that are involved in cardiotoxicity are not immediate early response genes.

## Potential limitations of our model

In this study we modelled cardiotoxicity by focusing on cardiomyocytes, the cell type ultimately affected by heart failure; however the heart is a complex organ consisting of multiple different cell types including endothelial cells, fibroblasts and pericytes [66] that likely interact during the course of drug treatment. Cell type interactions are therefore not captured in our system.

We generate iPSC-CMs through directed differentiation of iPSCs as it allows us to obtain cardiomyocytes from multiple individuals, and to use these cardiomyocytes in carefully controlled experiments where we can treat the same batch of cells with all cancer drugs. While these derived cells resemble ventricular cardiomyocytes, their phenotype is less mature than adult human cardiomyocytes. We try to mitigate this known limitation of iPSC-derived cell types by culturing our cells in a glucose-free, galactose-containing media that shifts the iPSC-CMs away from glycolysis as the primary metabolic pathway [67]. However, it is possible that our *in vitro* system may not fully recapitulate the *in vivo* situation.

While our iPSC-CMs are not proliferating during the course of our experiments, we note that they do express the cell cycle-related *TOP2A* gene in addition to *TOP2B* (log$_2$cpm *TOP2A* = 6.89, *TOP2B* = 7.85; *TOP2B*:*TOP2A* = 1.23) unlike left ventricle heart tissue (*TOP2A* = -0.8, *TOP2B* = 6.25 log$_2$cpm; *TOP2B*:*TOP2A* = 7.81), which is in line with these cells representing a more fetal state. It is therefore possible that some of the effects we observe in iPSC-CMs are mediated through inhibition of *TOP2A*. Indeed, we observe that TOP2i treatment significantly decreases both *TOP2B* and *TOP2A* mRNA expression.

For the bulk of our study, we selected a concentration of each drug that falls within the range of concentrations observed in patients being treated for cancer; however, the concentration of drug that the heart would be exposed to *in vivo* is hard to determine. Similarly, our single cell type *in vitro* approach cannot take into account the processing of these drugs that may occur in other organs *in vivo*, or other pharmacokinetic and pharmacodynamic features of the drugs that might differ between *in vivo* and *in vitro* environments.

There is controversy in the literature about the relative effects of different ACs and TOP2i on the heart [27–30]. In patient cohorts, comparisons across TOP2i are performed across individuals and studies. Our *in vitro* iPSC-CM system allowed us to systematically characterize the effects of multiple breast cancer drugs on cardiomyocytes in the same set of multiple individuals over time. Specifically, we profiled the response to DOX, EPI, DNR, MTX and TRZ following three and 24 hours of exposure in six individuals and identified a shared gene expression response signature across TOP2i that includes genes in known AC-induced cardiotoxicity loci. This work has potential clinical applications as it suggests that DOX, DNR and EPI affect heart cells similarly and that possible off-target effects on the heart may be induced following treatment with any one of the three ACs. We believe that the data and analysis presented here will be a resource for further studies into mechanistic and clinical effects of AC-induced cardiotoxicity.

## Materials and methods

### Ethics statement

iPSC lines from the iPSCORE resource were generated by Dr. Kelly A. Frazer at the University of California San Diego as part of the National Heart, Lung and Blood Institute Next

Generation Consortium [68]. The iPSC lines were generated with approval from the Institutional Review Boards of the University of California, San Diego and The Salk Institute (Project no. 110776ZF) and informed written consent of participants. Cell lines are available through the biorepository at WiCell Research Institute (Madison, WI, USA), or through contacting Dr. Kelly A. Frazer at the University of California, San Diego.

## Induced pluripotent stem cell lines

We used iPSC lines from six unrelated, healthy female donors of Asian ethnicity between the ages of 21 and 32 years with no previous history of cardiac disease or breast cancer from the iPSCORE resource [68]. The individuals are: Individual 1: UCSD129i-75-1 (iPSCORE_75_1, Asian-Irani, age 30), Individual 2: UCSD143i-87-1 (iPSCORE_87_1, Asian-Chinese, age 21), Individual 3: UCSD131i-77-1 (iPSCORE_77_1, Asian-Chinese, age 23), Individual 4: UCSD133i-79-1 (iPSCORE_79_1, Asian, age 24), Individual 5: UCSD132i-78-1 (iPSCORE_78_1, Asian-Chinese, age 21), and Individual 6: UCSD116i-71-1 (iPSCORE_71_1, Asian, age 32).

## iPSC culture

Cells were cultured at 37˚C, 5% $CO_2$ and atmospheric $O_2$. iPSCs were maintained in feeder-free conditions in mTESR1 (85850, Stem Cell Technology, Vancouver, BC, Canada) with 1% Penicillin/Streptomycin (30-002-Cl, Corning, Bedford, MA. USA) on 1:100 dilution of Matrigel hESC-qualified Matrix (354277, Corning). Cells were passaged using dissociation reagent (0.5 mM EDTA, 300 mm NaCl in PBS) every 3–5 days when the culture was ~ 70% confluent.

## Cardiomyocyte differentiation from iPSCs

Cardiomyocyte differentiation was performed as previously described [41]. Briefly, on Day 0, when a 10 cm plate of iPSCs reached 80–95% confluence, media was changed to Cardiomyocyte Differentiation Media (CDM) [500 mL RPMI 1640 (15-040-CM Corning), 10 mL B-27 minus insulin (A1895601, ThermoFisher Scientific, Waltham, MA) USA), 5 mL GlutaMAX (35050–061, ThermoFisher Scientific), and 5 mL of Penicillin/Streptomycin (100X) (30-002-Cl, Corning)] containing 1:100 dilution of Matrigel and 12 μM CHIR99021 trihydrochloride (4953, Tocris Bioscience, Bristol, UK). Twenty-four hours later (Day 1), the media was replaced with fresh CDM without CHIR99021. On Day 3, after 48 hours, spent media was replaced with fresh CDM containing 2 μM Wnt-C59 (5148, Tocris Bioscience). CDM was used to replace media on Days 5, 7, 10, and 12. Cardiomyocytes were purified through metabolic selection using glucose-free, lactate-containing media called Purification Media [500 mL RPMI without glucose (11879, ThermoFisher Scientific), 106.5 mg L-Ascorbic acid 2-phosphate sesquimagenesium (sc228390, Santa Cruz Biotechnology, Santa Cruz, CA, USA), 3.33 ml 75 mg/ ml Human Recombinant Albumin (A0237, Sigma-Aldrich, St Louis, MO, USA), 2.5 mL 1 M lactate in 1 M HEPES (L(+)Lactic acid sodium (L7022, Sigma-Aldrich), and 5 ml Penicillin/Streptomycin] on Days 14,16 and 18. On Day 20, purified cardiomyocytes were released from the culture plate using 0.05% trypsin/0.53 mM EDTA (MT25052CI, Corning) and counted using a Countess 2 machine. A total of 1.5 million cardiomyocytes were replated per well of a six-well plate, 400,000 cardiomyocytes per well of a 12-well plate and 55,000 cardiomyocytes per well of a 96-well plate in Cardiomyocyte Maintenance Media (500 mL DMEM without glucose (A14430-01, ThermoFisher Scientific), 50 mL FBS (MT35015CV, Corning), 990 mg Galactose (G5388, Sigma-Aldrich), 5 mL 100 mM sodium pyruvate (11360–070, ThermoFisher Scientific), 2.5 mL 1 M HEPES (H3375, Sigma-Aldrich), 5 mL 100X GlutaMAX (35050–061, ThermoFisher Scientific), and 5 mL Penicillin/Streptomycin). iPSC-CMs were

matured in culture for a further 7–10 days, with Cardiomyocyte Maintenance Media replaced on Days 23, 25, 27, 28, and 30.

## Cardiac Troponin T staining for iPSC-CM purity determination

After each iPSC differentiation, live cardiomyocyte purity was assessed using flow cytometry. Between Days 25–27, iPSC-CMs were detached from the plate using 0.05% trypsin/0.53 mM EDTA (MT25052CI, Corning) for 15 minutes. Trypsin was quenched with Cardiomyocyte Maintenance Media and cells were strained to remove clumps. One million cells were transferred into two test wells per sample and 3 control wells on a deep-well u-bottom plate (96 BRAND plates lipoGrade 96-Well Microplates (13-882-234, BrandTech Scientific, Essex, CT, USA). Test wells and the Zombie-only control wells were incubated with Zombie Violet Fixable Dye diluted in PBS (Zombie Violet Fixable Viability Kit (423113, BioLegend, San Diego, CA, USA) for 15 min at 4°C in the dark. After incubation, cells were rinsed 1X with PBS and 1X with autoMACS running buffer (MACS Separation Buffer (130-091-221, Miltenyi Biotec, San Diego, CA, USA). The cardiac Troponin T (TNNT2) antibody (Cardiac Troponin T Mouse, PE, Clone: 13–11, BD Mouse Monoclonal Antibody (564767, BD Biosciences, San Jose, CA, USA) was diluted in Permeabilization Buffer (FOXP3/Transcription Factor Staining Buffer Set, 00–5523, ThermoFisher Scientific) and added to both test wells and the TNNT2-only control well. Unstained control wells were resuspended in Permeabilization Buffer only. Cells were stained for 45 minutes at 4°C in the dark, then rinsed 2X in Permeabilization Buffer and resuspended in autoMACS buffer to be analyzed using flow cytometry. Ten thousand cells were analyzed per sample on a BD LSR Fortessa Cell Analyzer. To determine the percentage of live, TNNT2-positive cells, the following gating steps were taken: 1) The FSC versus SSC density plots were used to exclude cellular debris. 2) Cells were gated on FSC-H and FSC-A to exclude aggregate cells. 3) Violet laser-excitable cells were excluded as 'dead' cells. 4) Unstained iPSC-CMs were used as a negative-TNNT2 control to determine the TNNT2-positive range. Values reported are the mean of two technical replicates for each differentiation of each individual (n = 3).

## Drug stocks and usage

The panel of drugs used were Daunorubicin (30450, Sigma-Aldrich), Doxorubicin (D1515, Sigma-Aldrich), Epirubicin (E9406, Sigma-Aldrich), Mitoxantrone (M6545, Sigma-Aldrich) and Trastuzumab (HYP9907, MedChem Express). All drugs were dissolved in molecular biology grade water to a concentration of 10 mM per stock. DOX, DNR, EPI, and MTX stocks were stored at -80°C and working stocks used at 4°C for up to one week. TRZ was stored at a 1 mM concentration at 4°C for up to one month.

## Cell viability assay

Day 27 +/- 1 day iPSC-CMs were used for all drug treatments. iPSC-CMs were plated into three 96 well plates, excluding each plate's outermost rows. Eight concentrations [50 μM, 10 μM, 5 μM, 1 μM, 0.5 μM, 0.1 μM, 0.05 μM, 0.01 μM] were used for DOX, DNR, EPI, and MTX. The highest concentration (50 μM) was excluded for the TRZ treatments. The vehicle control (molecular biology-grade water) was used at the same volumes as the corresponding drug concentrations. Plate layouts were designed with the Well Plate Maker (wpm) package in R [69] to limit batch effects across plates. Each drug concentration was tested in quadruplicate per individual. iPSC-CMs were exposed to drug treatments for 48 hours. At the 48-hour time point, cell media was removed and stored at -80°C. Cells were washed two times with warm DPBS to remove any residual drug and dead cells, and cell viability was assessed using the

Presto Blue Cell Viability assay (A13262, Invitrogen) according to the manufacturer's instructions. Plate readings were obtained using a high throughput plate reader (Biotek Synergy H1) set to an excitation/emission of 460/490 nm. Data were processed following the manufacturer's instructions. Briefly, the background fluorescence was measured from wells containing no cells on each plate (n = 6). These values were averaged and subtracted from all wells of that plate, yielding a relative fluorescence unit (RFLU) value for each sample. Each RFLU value of the experiment was then normalized to the average RFLU of the vehicle at the same concentration (n = 4). This yielded a relative percent cell viability for each sample. The relative percent viability for each sample (n = 4) was used to generate a dose-response curve with the drc package in R [70]. $LD_{50}$ values, the concentration which killed 50% of the cardiomyocytes, for each drug on the panel was extracted from the model. Each individual had two independent dose-response curves performed from independent differentiations. The calculated $LD_{50}$ values for an individual were averaged from these two results to produce the reported $LD_{50}$ values. We were unable to derive meaningful $LD_{50}$ values for TRZ, due to lack of cell death observed at the treatment range used in this assay.

## Breast cancer cell line toxicity data

$EC_{50}$ data for breast cancer cell lines treated with drugs in our treatment panel were obtained from the DepMap Portal (https://depmap.org/portal/) using the PRISM Repurposing [33], CTRP $CTD^2$ [34,71], and Genomics of Drug Sensitivity in Cancer 2 (GDSC2) [35] databases. We selected those cell lines that had reported $EC_{50}$ values that fell within the ranges that each study tested (PRISM: 10 μM to 0.61 pM 8-point dilution series; CTRP: 66 μM to 0.65 μM, 16-point dilution series; GDSC2: 1 μM to 0.1 nM, 7-point dilution series). Cell lines are from invasive breast carcinomas (BT549, CAL51, HCC1143, HDQP1, MCF7, MDAMB231, MDAMB468, T47D, ZR751) and a breast ductal carcinoma *in situ* (HCC1806). DOX $EC_{50}$ values are reported as an average of $EC_{50}$ values from the PRISM and CTRP databases, EPI $EC_{50}$ values are an average from the PRISM and GDSC2 databases, and MTX and DNR $EC_{50}$ values are from the PRISM database.

## Human *in vivo* cancer drug concentration data

Clinically-measured human blood serum measurements were obtained from the literature using PubMed and the search terms "human plasma levels," or "pharmacokinetics in humans," and each chemotherapeutic drug by name (Doxorubicin, Epirubicin, Daunorubicin, Mitoxantrone, Trastuzumab). Each reference was reviewed to determine the number of individuals tested, cancer diagnosis, number of treatment cycles, and cumulative dosage for each drug. All studies obtained which did not include plasma levels for at least one human, or the term pharmacokinetics were excluded. We collected the lowest and highest serum concentration measurements taken closest to one hour after infusion, and values were transformed to μM, when needed. If the study did not provide the maximum and minimum concentration, the median or mean +/- standard deviation, maximum or minimum is reported. The final set of reported studies are included in S3 Table [35,72–95].

## Lactate dehydrogenase activity assay

Lactate dehydrogenase activity (LDH) was measured from 5 μL cell culture media using the Lactate Dehydrogenase Activity Assay Kit (MAK066, MilliporeSigma) according to the manufacturer's instructions. Each sample was assayed in triplicate. LDH activity was measured as the change in absorbance of the sample relative to the change in absorbance of the media background control, calculated relative to a standard curve, before and after incubation for 10 min

at 37˚C. This value was normalized to the drug concentration-specific VEH sample value for each individual.

## Treatments for gene expression measurements

1.5 million iPSC-CMs were plated in 6-well plates. Between Days 27–29, iPSC-CMs were treated with 0.5 µM DNR, DOX, EPI, MTX, TRZ, or vehicle in fresh cardiomyocyte maintenance media. iPSC-CMs were collected three and 24 hours post-treatment, resulting in 72 samples from 6 individuals. iPSC-CMs were washed twice with ice-cold PBS and manually scraped in cold PBS on ice. Cell pellets were flash-frozen and stored at −80˚C.

## Calcium imaging and analysis

Day 20 iPSC-CMs from three individuals (Individuals 2, 3, and 5) were plated on Matrigel-coated 12-well plates, at a density of 400,000 cells/well. Between Days 27–30, the cells were subjected to a 24-hour treatment with 0.5 µM of each drug. Fluorescence measurements of cytosolic calcium were obtained using the Fluo-4 AM probe (F14201, Invitrogen, Waltham, MA, USA), with the protocol as specified by the manufacturer. The probe was prepared in DMSO and subsequently applied to each well to achieve an 8 µM final concentration. Cells were incubated at 37˚C in the dark for 25 min, rinsed twice with Hank's Balanced Salt Solution (HBSS; 14025092, ThermoFisher) and protected from light. Imaging of Fluo-4 AM-treated iPSC-CMs was performed at 37˚C on an Olympus spinning disc confocal microscope, using a 488 nm excitation wavelength at 20% power. Fluorescence intensity values were captured at a frame rate of 31.34 frames per second for approximately 10 seconds.

Calcium transient recordings were archived as .avi files and processed with CALIMA [96] to obtain 500–1,000 regions of interest for calcium transient recordings. The resultant CALIMA output was further processed using Clampfit (Axon pCLAMP 10 Electrophysiology Data Acquisition & Analysis Software, Molecular Devices), employing Gaussian curve fitting to analyze the calcium transients, which enabled the computation of parameters such as amplitude, rising slope, decay slope, and full-width-at-half-maximum for each calcium transient within the recording space. This process was carried out for each drug treatment in each individual.

Individual peaks from the CALIMA output were subsequently analyzed in R. A representative trace from the ensemble of all measured traces per individual was computed to represent the amplitude of the calcium signal, termed "Intensity". These average traces were then utilized to detect calcium transient peaks following signal noise removal. Peaks were defined as instances where Intensity[i] > (Intensity[i + 1] & Intensity[i—1]) & (Intensity[i] > quantile (Intensity, 0.6)). Beat rate was calculated by dividing the total number of detected peaks by the duration of the recording.

For PCA clustering analysis, Mean Amplitude, Rising Slope, Decay Slope, Contraction Rate, and Full-Width-at-Half-Maximum parameters were averaged across individuals for each drug condition.

## RNA extraction

RNA was extracted from the flash-frozen cell pellets using the Zymo dual DNA/RNA extraction kit (D7001, Zymo, Irvine, CA, USA). Extractions were performed in batches of 12 where all treatments and timepoints per individual were extracted in the same batch. RNA concentration and quality was measured using the Agilent 2100 Bioanalyzer. RIN scores were greater than 7.5 for all samples with a median of 9.35, 9.6, 9.3, 9.6, 9.65, and 9.55 across treatments for individuals 1–6, respectively.

### RNA-seq library preparation

RNA-seq libraries were generated using 250 ng of RNA using the NEBNext Poly(A) mRNA Magnetic Isolation Module kit (E7490L, Ipswich, MA, USA), NEBNext Ultra II RNA library prep with Sample Purification Beads kit (E7775K) and the NEBNext Multiplex Oligos for Illumina (96 Unique Dual Index Primer Pairs) kit (E6440S). Libraries were prepared in time and treatment balanced batches of 12 for each individual, following the manufacturer's protocol. RNA-seq library sizes were determined using the Agilent 2100 Bioanalyzer DNA chip before quantification and sequencing on the NextSeq 550 using 75 bp single-end reads. All samples (n = 72) were pooled and sequenced together on four lanes for a total of five runs to generate a minimum of 20 million reads per sample (median = 31,666,884).

### RNA-seq analysis

Reads were assessed for quality using the MultiQC package [97] in Linux. Subread [98,99] was used to align the reads to the hg38 reference genome, with the featureCounts function within the package used to quantify the read number across annotated genes in R. We transformed the counts to $\log_2$ counts per million with the edgeR package [100] and excluded genes with a mean $\log_2$ cpm < 0 across samples, leaving 14,084 expressed genes for downstream analysis.

### Differential expression analysis

We performed pairwise differential expression analysis using the edgeR-voom-limma pipeline [101], contrasting each treatment against the vehicle at each timepoint. DE genes are defined as those genes for each treatment-vehicle pair that meet an adjusted $P$ value threshold of < 0.05.

   We jointly modeled pairs of tests with the Cormotif package in R [102]. Cormotif implements a Bayesian clustering approach that identifies common expression patterns (or correlation motifs) that best fit the given data. We used TMM-normalized $\log_2$ cpm values as input and paired each drug treatment with the VEH at the corresponding timepoint. Using the BIC and AIC, we found that the best fit model to the data was four motifs. A gene was considered to belong to one of the four motifs when it had > 0.5 probability of belonging to the motif and < 0.5 probability of being in any of the other motifs. This threshold yields assignment of 99.6% of genes to a single motif.

### Expression variance analysis

Gene expression variance was analyzed using the $\log_2$ cpm counts in R as has been previously described [55]. The mean and variance were calculated by treatment (n = 6) and time (n = 2) across individuals for each expressed gene. The function var.test was used to assess the variance of each gene of each treatment compared to vehicle at each timepoint.

   To assess gene expression variance following AC treatment in a larger number of individuals, we obtained gene expression measurements for vehicle and 0.625 μM DOX-treated iPSC-CMs from 45 individuals from Knowles et al. [15].

### Gene ontology and pathway analysis

Gene set enrichment analysis was performed on each gene set and a background set of all expressed genes using the gProfiler2 tool in R [103,104]. Significance of pathway enrichment was determined at an FDR < 0.05.

## Comparison with published data

*AC-sensitive chromatin regulators.* A list of 408 curated chromatin regulator genes and 54 AC-sensitive chromatin regulator genes were obtained from Seoane *et al.* [22]. We intersected these sets with our response categories to determine the enrichment of chromatin regulators within the early-acute, early-sustained, and late response sets with respect to the no-response set. The chi-square test of proportions was used to test for enrichment. A $P < 0.05$ was considered a significant difference in proportions, and the transformed $-\log_{10} P$ values are reported as heatmaps using ComplexHeatmap [105,106] in R.

*eGenes in heart.* We obtained a list of 9,642 eGenes (q value $< 0.05$) from the GTEx v8 human heart:left ventricle data (http://www.gtexportal.org). We intersected this list with our expressed genes (14,084) to obtain a list of 6,261 expressed eGenes and 7,823 expressed genes that are not eGenes (not eGene category). We then tested for enrichment of proportions of DE and not DE between the eGene and not eGene sets using the chi-square test of proportions. $P < 0.05$ was considered a significant difference in proportions.

*DOX response eGenes.* A list of 518 marginal effect eQTLs from Knowles *et al.*, and 376 DOX-response eQTLs, were used to intersect with our data [15]. The two eGene sets shared 104 genes, which were excluded from our analysis, leaving 417 genes baseline eGenes, and 273 DOX-response eGenes. A chi-square test was used to test for enrichment of DE genes found in baseline eGenes and DOX-response eGenes by time and treatment. $P < 0.05$ was considered a significant difference in proportions.

*Genes in AC-induced cardiotoxicity GWAS loci.* To investigate the drug-induced gene expression response in cardiotoxicity-associated loci, we used data from three GWAS-based studies. First, we selected the 50 SNPs most associated with cardiotoxicity in a cohort of ~3,000 breast cancer patients treated with an anthracycline [8]. The list of SNPs was converted to .bed format and the closest gene transcription start site to each SNP identified using BEDtools [107]. We removed genes that were not expressed in our data. We also used the GTEx eQTL database (https://gtexportal.org) to associate each SNP with genes in any tissue. We filtered out redundant gene names and those genes not expressed in our data. We combined the closest gene set and eQTL set to generate a list of 30 genes to interrogate from this study.

Second, we included three SNPs associated with AC-induced cardiotoxicity (rs2229774, rs7853758, rs17863783), which are currently recommend for testing in pediatric patients by the Canadian Pharmacogenomics Network for Drug Safety [42]. Following the same pipeline described above, these SNPs, yielded a set of four genes.

Third, we included genes prioritized for their involvement in AC-induced cardiotoxicity by TWAS [43] and filtered out genes not expressed in our data leaving a set of four genes.

We combined genes identified from all three studies for interrogation of the log fold change between drug treatment and vehicle at three and 24 hours.

## Supporting information

**S1 Fig. Cardiomyocytes can be generated at high purity across six individuals. (A)** Representative image of flow cytometry data indicating the proportion of TNNT2 positive cells in one differentiation experiment for each individual based on the fluorescent intensity of the phycoerythrin-labeled TNNT2 antibody and a sample of unlabeled iPSC-CMs (red cell population). **(B)** Percentage of cells that are positive for expression of TNNT2 for each individual. Data representative of three independent differentiation experiments used for the two drug dose-response curves, and RNA collection. The dashed line represents high-purity iPSC-CMs ($> 70\%$ TNNT2 positive).
(TIF)

**S2 Fig. Dose-response curves are reproducible across replicate cardiomyocyte differentiations from the same individual.** Proportion of viable cardiomyocytes following exposure to increasing concentrations of each drug. Cell viability in replicate one (solid line) and two (dashed line) in Individual three (orange), and Individual five (green) was assessed following 48 hours of drug treatment. Viability was determined at each drug concentration in quadruplicate, and the mean value was selected for generation of the dose-response curves using a four-point log-logistic regression with the upper asymptote set to one. Shading represents the 95% confidence interval from the regression analysis for Individual three (light orange) and Individual five (light green).
(TIF)

**S3 Fig. Cancer drugs that decrease cardiomyocyte viability induce cellular stress.** Pearson correlation between cardiomyocyte viability following drug treatment at eight different concentrations for 48 hours, and the level of lactate dehydrogenase released into the cell culture media across individuals. Data points are colored by individual (1,2,3,4,5,6).
(TIF)

**S4 Fig. Treatment with ACs at a dose of 0.5 μM for 48 hours induces effects on cardiomyocyte viability.** Proportion of viable cells following treatment with each drug (DOX: mauve; EPI: pink; DNR: yellow; MTX: blue; TRZ: dark green; VEH: light green) in each individual (1,2,3,4,5,6) at five sub-micromolar drug concentrations.
(TIF)

**S5 Fig. RNA-seq sample quality is equivalent across individuals, treatments, and time points. (A)** RNA integrity score for each sample categorized by drug type and drug treatment time. Data inclusive of six individuals. **(B)** Total number of RNA-sequencing reads categorized by treatment type (DOX: mauve; EPI: pink; DNR: yellow; MTX: blue; TRZ: dark green; VEH: light green). Each drug treatment category includes data from six individuals across two time points. **(C)** Total number of RNA-seq reads for each of the 72 samples. Each sample is denoted by drug.individual.timepoint.
(TIF)

**S6 Fig. RNA-seq samples cluster by treatment type, timepoint, and individual.** Pearson correlation of $\log_2$ cpm values across all pairs of samples.
(TIF)

**S7 Fig. PC1 associates with drug treatment and treatment time, while PC2 associates with individual.** Demonstration of variance contributed to the first two principal components from three major covariates in the study: individual, treatment, and time. **(A)** Variance of individual as a function of PC1 and PC2. The correlation between individual and each PC is calculated using a linear model. *P* values represent the significance of the F-statistic from the model. **(B)** Variance of treatment as a function of PC1 and PC2. **(C)** Variance of time as a function of PC1 and PC2.
(TIF)

**S8 Fig. Thousands of gene expression changes are induced in response to TOP2i treatment over 24 hours.** Volcano plots representing genes that are differentially expressed between drug and vehicle treatment at each timepoint. Genes that are significantly up-regulated in response to treatment (adjusted *P* value < 0.05) are represented in blue, and genes that are significantly down-regulated are represented in red. The number of genes that are up- and down-regulated

is given for each plot.
(TIF)

**S9 Fig. ACs affect expression of nearly half of all expressed genes after 24 hours of treatment. (A)** Percentage of genes that are differentially expressed between each drug treatment and the vehicle following three and 24 hours of treatment. **(B)** Log$_2$ fold change between drug-treated and VEH-treated samples for all 14,084 expressed genes following three and 24 hours of treatment.
(TIF)

**S10 Fig. A small number of genes respond to a single drug only. (A)** Distribution of adjusted *P* values for all genes classified as drug-specific response genes based on overlap of significantly differentially expressed genes meeting the adjusted *P* value cutoff of 0.05 across drugs at three and 24 hours. *P* values are shown for all drug treatments for the set of drug-specific genes (DOX: mauve; EPI: pink; DNR: yellow; MTX: blue; TRZ: dark green; VEH: light green). **(B)** The log$_2$ fold change of all genes meeting a stringent adjusted *P* value cutoff of 0.01 for the drug of interest & $>$ 0.05 for all other drugs to identify drug-specific response genes at three and 24 hours. **(C)** Examples of expression levels of stringently-identified drug-specific response genes across drug treatments at each time point.
(TIF)

**S11 Fig. Stringently-identified drug-specific response genes are enriched in biological processes. (A)** Biological processes enriched amongst genes classified as stringent DNR-specific and stringent MTX-specific response genes compared to all expressed genes following three hours of treatment. The top ten most enriched biological processes from Gene Ontology analysis that meet an adjusted *P* value cutoff of 0.05 are shown except for MTX-stringent where only processes to the right of the dashed line are significantly enriched. Dot size represents the number of stringent drug-specific response genes that are annotated as belonging to the particular biological process. There are no DOX-specific or EPI-specific response genes that pass the stringent threshold at three hours. **(B)** Biological processes enriched amongst genes classified as stringent DOX-specific and stringent MTX-specific response genes compared to all expressed genes following 24 hours of treatment. There are no EPI-specific or DNR-specific response genes that pass the stringent threshold at 24 hours.
(TIF)

**S12 Fig. Nominally significant TRZ response genes show enrichment in cancer-related pathways. (A)** Number of TRZ response genes that pass a nominal p-value cutoff (unadjusted *P* $<$ 0.05) at three and 24 hours. **(B)** Expression of *PHLDA1*, a three hour TRZ response gene, and *ANKRD2*, a 24 hour TRZ response gene. (TRZ: dark green; VEH: light green). **(C)** Top KEGG pathways represented amongst TRZ response genes at three and 24 hours. Dashed black line is -log$_{10}$ *P* $<$ 0.05.
(TIF)

**S13 Fig. Four gene expression signatures capture the response to TOP2i over time. (A)** Bayesian information criterion (BIC) and Akaike information criterion (AIC) at increasing numbers of Cormotif correlation motifs following joint modeling of pairs of tests. **(B)** Gene expression levels of genes assigned to each TOP2i response signature in each drug treatment at each time point. The *AHDC1* gene represents the Early-acute response motif (red), the *MEX3A* gene represents the Early-sustained response motif (blue), the *MED29* gene represents the Late response motif (green), and *GCNT1* represents the No response motif (purple). **(C)** The top ten most enriched biological processes (adjusted *P* value $<$ 0.05) that are enriched in

the response gene categories compared to all expressed genes. Dot size represents the number of correlation motif genes that are annotated as belonging to the particular biological process.
(TIF)

**S14 Fig. Gene expression variance across 45 individuals increases in iPSC-CMs treated with DOX. (A)** Mean of 12,317 expressed genes in untreated and 0.625 μM DOX-treated samples [15]. **(B)** Variance of gene expression across 45 individuals in untreated and 0.625 μM DOX-treated samples. Asterisk indicates $P < 0.001$.
(TIF)

**S15 Fig. Replication of gene expression response in AC-induced cardiotoxicity loci.** Gene expression levels of DOX-responsive genes in cardiotoxicity-associated loci in our data (VEH: light green; DOX: mauve; EPI: pink; DNR: yellow; MTX: blue; TRZ: dark green), and in DOX-treated samples across 45 individuals (untreated: lime and 0.625 μM DOX: purple) [15].
(TIF)

**S1 Table. Average viability for each dose response curve analysis.** Data related to Figs 1B, S2, S3, and S4.
(XLSX)

**S2 Table. Estimated LD$_{50}$ using average viability from both replicates and estimated LD$_{50}$ for each individual replicate.** Data related to Fig 1C.
(XLSX)

**S3 Table. Drug concentrations in cancer patient serum based on literature.** Data related to Fig 1D.
(XLSX)

**S4 Table. Table of relative LDH values.** Data related to S3 Fig.
(XLSX)

**S5 Table. Measurements from calcium experiments.** Data related to Fig 2B–2F.
(XLSX)

**S6 Table. RNA-seq sample metadata.** Data related to S1 and S5 Figs.
(XLSX)

**S7 Table. Pairwise differential expression analysis for DNR vs VEH.** Data related to Figs 3B–3D, 7, S8, S9, and S10.
(XLSX)

**S8 Table. Pairwise differential expression analysis for DOX vs VEH.** Data related to Figs 3B–3D, 7, S8, S9, and S10.
(XLSX)

**S9 Table. Pairwise differential expression analysis for EPI vs VEH.** Data related to Figs 3B–3D, 7, S8, S9, and S10.
(XLSX)

**S10 Table. Pairwise differential expression analysis for MTX vs VEH.** Data related to Figs 3B–3D, 7, S8, S9, and S10.
(XLSX)

**S11 Table. Pairwise differential expression analysis for TRZ vs VEH.** Data related to Figs 3B–3D, 7, S8, S9, S10, and S12.
(XLSX)

**S12 Table. Stringently-identified response genes for each treatment.** Data related to S10 and S11 Figs.
(XLSX)

**S13 Table. Results of Cormotif analysis.** Data related to Figs 4A–4C, 7, and S13.
(XLSX)

**S14 Table. List of genes assigned to each motif.** Data related to Figs 4A–4C, 7, and S13.
(XLSX)

**S15 Table. Calculated mean and variance of log$_2$ cpm values.** Data related to Fig 5A–5C.
(XLSX)

# Acknowledgments

We thank all members of the Ward Lab for helpful discussions. We thank Dr. Kelly A. Frazer and the University of California San Diego for providing the iPSC lines through the iPSCORE resource. We thank the Molecular Genomics Core Facility at the University of Texas Medical Branch for sequencing the RNA-seq libraries, the Flow Cytometry and Cell Sorting Core Facility for access to flow cytometers, and Dr. Michael Sheetz and the Mechanomedicine Microscopy facility for access to the spinning disc confocal microscope. The authors acknowledge the Texas Advanced Computing Center (TACC) at The University of Texas at Austin for providing HPC resources that have contributed to the research results reported within this paper.

# Author Contributions

**Conceptualization:** Michelle C. Ward.

**Formal analysis:** E. Renee Matthews, Omar D. Johnson, Michelle C. Ward.

**Funding acquisition:** Michelle C. Ward.

**Investigation:** E. Renee Matthews, Omar D. Johnson, Kandace J. Horn, José A. Gutiérrez, Simon R. Powell.

**Project administration:** Michelle C. Ward.

**Supervision:** Michelle C. Ward.

**Visualization:** E. Renee Matthews.

**Writing – original draft:** E. Renee Matthews, Michelle C. Ward.

**Writing – review & editing:** Omar D. Johnson, José A. Gutiérrez, Simon R. Powell.

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
