## [Decision Letter · Decision Letter 0]

26 Oct 2023

Dear Dr Ward,

Thank you very much for submitting your Research Article entitled 'Anthracycline-induced cardiotoxicity associates with a shared gene expression response signature to TOP2-inhibiting breast cancer drugs in cardiomyocytes' to PLOS Genetics.

The manuscript was fully evaluated at the editorial level and by independent peer reviewers. The reviewers appreciated the attention to an important problem, but raised some substantial concerns about the current manuscript. Based on the reviews, we will not be able to accept this version of the manuscript, but we would be willing to review a much-revised version. We cannot, of course, promise publication at that time.

If you decide to revise the manuscript for further consideration at PLOS Genetics, please aim to resubmit within the next 60 days, unless it will take extra time to address the concerns of the reviewers, in which case we would appreciate an expected resubmission date by email to plosgenetics@plos.org.

We are sorry that we cannot be more positive about your manuscript at this stage. Please do not hesitate to contact us if you have any concerns or questions.

Yours sincerely,

Anna Di Rienzo

Academic Editor

PLOS Genetics

David Kwiatkowski

Section Editor

PLOS Genetics

The editors would be willing to consider a substantially revised manuscript. While all reviewers' points need careful attention, the revised manuscript should clearly address the following points:

1) a citation to the paper by Reyes et al 2018, as discussed by Reviewer 3, and a discussion of how the previous findings corroborate or illuminate the results reported in this manuscript, despite the obvious differences in sample size (1 vs 6), the larger number and types of drug used in this manuscript, etc.

2) a discussion of the observed lack of transcriptional response to TRZ and its implications for the experimental system and for the overall results.

Reviewer's Responses to Questions

**Comments to the Authors:**

Reviewer #1: General summary.

Cardiotoxicity is a common side effect of treatment with anthracyclines, a common class of chemotherapeutic. However, variation in cardiotoxicity caused by different types of anthracyclines is not well understood, nor are the genetic underpinning of inter-individual variation in this side effect.In this study, the authors have treated 6 iPSC-derived cardiomyocyte cell lines with various anthracyclines, and documented the gene expression changes that occur, demonstrating that changes occur in genes near SNPs previously associated with inter-individual variation in AC-induced cardiotoxicity. Overall, this is a solid piece of work, adding an interesting layer to the known body of knowledge on AC-induce cardiotoxicity.

Minor concerns.

- At first glace the title of the paper is a bit difficult to understand/digest. I would suggest rethinking it (I am not insisting on this).

- Are the induced cardiomyocytes proliferative? TOP2A in general would only be expected to be expressed during the cell cycle, thus, if these in vitro cardiomyocytes are proliferating, and cardiomyocytes in vivo are not proliferating, the general results may skew towards a misleadingly large involvement of TOP2A in cardiotoxicity. Consistent with this TOP2A expression is far higher in the in vitro cardiomyocyte system than in left ventricle heart tissue (per the authors own result, line 252; seems to be a 60 fold difference…). TOP2A may be playing no role at all in vivo (as it is likely not expressed). This may be worth commenting upon. Perhaps the in vitro system would be more faithful if the cells were arrested (I’m not saying the authors need to do that here).

- Some of the conclusions could benefit from being a bit more specific, e.g. line 419-420 “These results imply that treatment induced variability in expression across individuals in a subset of genes” comes across as somewhat weak. The paper may actually benefit from removing some of the less convincing analyses (i.e. this could easily be a 5-6 figure paper without losing much).

- It would likely be worth highlighting the utility of the data generated as a resources for other investigators who are interested in studying AC included cardiotoxicity. This seems like a major selling point of the data that I don’t think has been emphasized.

Reviewer #2: This study by Matthews et al identified that the gene expression signature of anthracycline-induced cardiotoxicity is similar to responses by TOP2-inhibiting (TOP2i) breast cancer drugs in iPSC-derived cardiomyocytes (iPSC-CMs). The authors studied anthracycline drugs vs trastuzumab on iPSC-CMs from six currently healthy females and found that all TOP2i drugs caused cell death at concentrations measured in patient serum while trastuzumab (TRZ) does not. A sub-lethal dose of TOP2i treatment caused changes in iPSC-CM calcium handling and induced thousands of gene expression changes. The early changes are enriched in chromatin regulators and varies between each individual. The authors then generated an DOX response eQTL and show correlation of expression of eight genes, including RARG and SLC28A3, that were reported to associate with cardiotoxicity by GWAS or TWAS. This study demonstrates the utility of iPSC-CMs as a way to correlate clinical phenotype with changes in gene expression perturbation on a genome-wide level and helps to clarify the mechanism of TOP2i-induced cardiotoxicity for future therapy development. Some suggestions for improvement are listed below:

1. Figure 9D – The correlation seems weak and heterogenous between toxicity score and the changes in expression of genes that have been shown to be correlated with cardiotoxicity in individuals from GWAS study. To verify that the gene expression changes are meaningful for the toxicity effects, it would be helpful to perform gain- or loss-of-function studies for these genes (TNS2, SLC28A3, RM1) in TOP2i-treated iPSC-CMs and measure the level of troponin/LDH release to see the direction of contribution of these genes to the disease phenotype.

2. The data presented for gene expression changes and cardiotoxicity effects does not specifically describe whether there exists a positive/negative correlation within the six individual iPSC-CM lines but as an average of the change in all six individual iPSC-CMs. It would help to have more delineation of the phenotype of the six individuals and the data from their iPSC-CMs. While the iPSC lines are said to be from healthy female donors, do any of these individuals have a history of or risk factor for breast cancer? Have these individuals been treated with TOP2i clinically and develop or not cardiotoxicity? Is it known what their genotypes are for the GWAS-associated gene loci? From the current assays, are there any gene expression changes that tracks with the toxicity phenotype across these six individuals (e.g. highest change in gene expression correlates highest LDH/Troponin release within the six iPSC-CM lines sampled regardless of whether these are GWAS-associated genes?)

3. Given the large amount of gene expression data presented from many individual iPSC lines treated with many TOP2i drugs, the main message of this study can get somewhat loss with all the different gene changes with different drugs. It would be helpful if the authors can focus on a specific message (i.e. changes in X gene expression correlate the best with increase TOP2i toxicity in iPSC-CMs where X can be chromatin regulating, base excision, or GWAS associate genes, etc).

Reviewer #3: In this manuscript, Renee et al. investigated the effect of 4 TOP2 inhibitors and trastuzumab on cell viability, calcium handling and gene expression in iPSC derived cardiomyocytes obtained from six healthy female individuals. The author first studied effect of TOP2i on cell death, calcium dysregulation, gene expression, and investigated the reasons of individual variation. Although a lot of results have been reported from this study, there is limited novel conclusion that could derived from this work. The transcriptome changes that induced by Dox have been studied in iPSC derived cardiomyocytes (10.1016/j.taap.2018.07.020). The manuscript summarized the general level of transcriptomic signatures from TOP2i but does not provide either molecular mechanism investigations or translational impacts. Moreover, the abstract indicated that the main goal of this study is to investigate whether all TOP2i would behave the same way regarding their induction of cardiotoxicity. The presented results show some differences; however, in the absence of a true negative control (see detailed comment below), it is difficult to interpret what these observed differences mean. Therefore, for the most part, the current study failed to address the question it set out to address.

Comments to authors:

1. Study design issues:

a. The study of drug induced gene expression changes were conducted at 2 preselected time points and at a fixed concentration of 5 drugs. The authors then try to categorize expression changes across time points into four clusters. Pharmacology of drug response is a dynamic phenomenon with time and concentration highly intertwined. It is difficult to justify the meaning of this kind of arbitrary classification in real world.

2. Technical issues:

a. Trastuzumab (TRZ) is tested in this study as a drug that do not target TOP2. However, TRZ is also known to lead to cardiotoxicity. It is not clear why the investigators did not observe any cell death after exposing iPSC-CM cells to this drug. It is even more bizarre to see no gene expression change after TRZ treatment when compared to vehicle control. Is that because wrong dose was used, solubility issue…? The lack of expression impact is especially worrisome. The authors need to add plausible explanation to the discussion, so readers won’t question the validity of the model/work. Furthermore, because this out of class drug is not working in the experimental system, the authors literally just compare phenotypic differences among TOP2is. When compare among items, there will always be differences observed. The authors presented these differences, but was not able to draw meaningful conclusions with the data.

Additional comments:

1. The variations are large across 6 individuals. Repeated measurement of gene expression could be helpful in validating the results.

2. The transcriptomic changes upon Dox intervention have been studied in iPSC cardiomyocyte. The author has generated new data from other TOP2i but did not further investigate the findings. It would be interesting and critical to know what drug specific transcriptomic changes are.

3. The impact of findings from this manuscript is not clear. It is critical to know whether any of these genes have clinical impacts. The author could elaborate on this in the discussion.

4. Do any gene candidates from DE could affect the drug effectiveness on iPSC-cardiomyocytes?

5. Why only 3 individual’s iPSC were selected for Calcium investigation? Especially, these three (individual 2, 3, 5) have distinct LD50 in Dox group in Figure 1C.

6. Figure 4A, it is not clear that TRZ treatment fall into a cluster. Especially the correlation between TRZ 3h and TRZ 24h is relatively low (0.16).

7. What is the potential mechanism behind the fact that HSV showing up in Figure 4D?

8. For Figure 7A and 7B, could you use PCA plot (or similar plot) to show the similarities/dissimilarities of different treatment groups? It is not clear by just showing the variation in gene expression.

9. lines 406-408, referring to fig 7C, the manuscript stated “…at 24 hours it clustered based on whether the drugs are ACs or not”. However, it looks like all five drugs cluster together in Fig 7C. Please also show the vehicle result in this plot as well.

10. No statistical tests on Figure 3B, 6B, 8D, and 9D.

11. It is very hard to understand and interpret Figure 4A. What exactly does the correlation done on? Gene expression (1 gene or all genes) or drug response (like LD50).

12. The text is opposite to what Figure 9D shows.

**Have all data underlying the figures and results presented in the manuscript been provided?**

Reviewer #1: Yes

Reviewer #2: Yes

Reviewer #3: None

PLOS authors have the option to publish the peer review history of their article (what does this mean?). If published, this will include your full peer review and any attached files.

Reviewer #1: No

Reviewer #2: **Yes: **Sean Wu

Reviewer #3: No

---

## [Editor Report · Decision Letter 1]

31 Jan 2024

Dear Dr Ward,

We are pleased to inform you that your manuscript entitled "Anthracyclines induce cardiotoxicity through a shared gene expression response signature" has been editorially accepted for publication in PLOS Genetics. Congratulations!

Yours sincerely,

Anna Di Rienzo

Academic Editor

PLOS Genetics

David Kwiatkowski

Section Editor

PLOS Genetics

Comments from the reviewers (if applicable):

**Data Deposition**

http://datadryad.org/submit?journalID=pgenetics&manu=PGENETICS-D-23-01098R1

**Press Queries**

---

## [Editor Report · Acceptance letter]

22 Feb 2024

PGENETICS-D-23-01098R1 

Anthracyclines induce cardiotoxicity through a shared gene expression response signature 

Dear Dr Ward, 

We are pleased to inform you that your manuscript entitled "Anthracyclines induce cardiotoxicity through a shared gene expression response signature" has been formally accepted for publication in PLOS Genetics! Your manuscript is now with our production department and you will be notified of the publication date in due course.

With kind regards,

Bernadett Koltai

PLOS Genetics

On behalf of:
